# A Tale of Two Graphs: Separating Knowledge Exploration from Outline Structure for Open-Ended Deep Research

**Zhuofan Shi** [* ‡ 1]  **Ming Ma** [* ‡ 2]  **Zekun Yao** [‡ 3]  **Fangkai Yang** [† 1]  **Jue Zhang** [† 1]  **Dongge Han** [1]  **Victor Rühle** [1]
**Qingwei Lin** [1]  **Saravan Rajmohan** [1]  **Dongmei Zhang** [1]

## Abstract

Open-Ended Deep Research (OEDR) pushes LLM agents beyond short-form QA toward long-horizon workflows that iteratively search, connect, and synthesize evidence into structured reports. However, existing OEDR agents largely follow either linear "search-then-generate" accumulation or outline-centric planning. The former suffers from lost-in-the-middle failures as evidence grows, while the latter relies on the LLM to implicitly infer knowledge gaps from the outline alone, providing weak supervision for identifying missing relations and triggering targeted exploration. We present DualGraph memory, an architecture that separates *what* the agent knows from *how* it writes. DualGraph maintains two co-evolving graphs: an Outline Graph (OG), and a Knowledge Graph (KG), a semantic memory that stores fine-grained knowledge units, including core entities, concepts, and their relations. By analyzing the KG topology together with structural signals from the OG, DualGraph generates targeted search queries, enabling more efficient and comprehensive iterative knowledge-driven exploration and refinement. Across four established OEDR benchmarks, DualGraph consistently outperforms state-of-the-art baselines in report depth, breadth, and factual grounding; for example, it reaches a 53.08 RACE score on DeepResearch Bench with GPT-5. Moreover, ablation studies confirm the central role of the dual-graph design.

[*]Equal contribution  [†]Corresponding authors.  [‡]Work done during an internship at Microsoft. [1]Microsoft [2]Institute of Neuroscience, Center for Excellence in Brain Science and Intelligence Technology, Chinese Academy of Sciences [3]School of Computer Science and Engineering, South China University of Technology. Correspondence to: Fangkai Yang <fangkaiyang@microsoft.com>, Jue Zhang <juezhang@microsoft.com>.

*Proceedings of the 43rd International Conference on Machine Learning*, Seoul, South Korea. PMLR 306, 2026. Copyright 2026 by the author(s).

## 1. Introduction

The rapid evolution of Large Language Models (LLMs) has expanded their real-world applications beyond short-form question answering toward Open-Ended Deep Research (OEDR) (Zhang et al., 2025; Li et al., 2026b), which typically requires agents to navigate vast, heterogeneous information spaces to synthesize comprehensive reports on complex topics. Recent benchmarks, such as DeepResearch Bench (Du et al., 2025), DeepResearch Bench II (Li et al., 2026a), DeepResearchGym (Coelho et al., 2025), and DeepConsult (youdotcom-oss, 2025), underscore the difficulty of OEDR tasks, revealing that even frontier models struggle to systematically explore information spaces, identify missing or under-researched knowledge and relations, and maintain factual grounding over long-horizon research workflows.

Current approaches to OEDR predominantly fall into two paradigms. The first, linear accumulation (Han et al., 2025; LangChain; Felovic), employs a "search-then-generate" strategy (Li et al., 2025b; Jin et al., 2025) that aggregates documents into a long context window for single-pass synthesis. Although simple, this approach often suffers from "lost-in-the-middle" phenomena (Wang et al., 2024) as the volume of evidence grows (Qiao et al., 2025). The second, more sophisticated paradigm involves outline-centric agents, exemplified by systems like STORM (Shao et al., 2024), WebWeaver (Li et al., 2026b), and DeepScholar (Patel et al., 2025). These systems generate a hierarchical outline-like memory to guide iterative search and evidence collection for structuring the final report. While outlining improves structural coherence, it is often treated as a proxy for the agent's epistemic state, forcing the LLM to infer missing knowledge from sparse section headers, and it could fail to trigger the in-depth and cross-section searches for completeness, and does not explicitly identify which concepts or relations remain unexplored. Moreover, shared knowledge is easily duplicated or scattered across multiple outline sections, making it difficult to maintain a single source of truth for contradiction detection, global updates, and reasoning across various sections. For example, a technical concept like "attention mechanism" may be foundational to both the background and methodology sections. This motivates

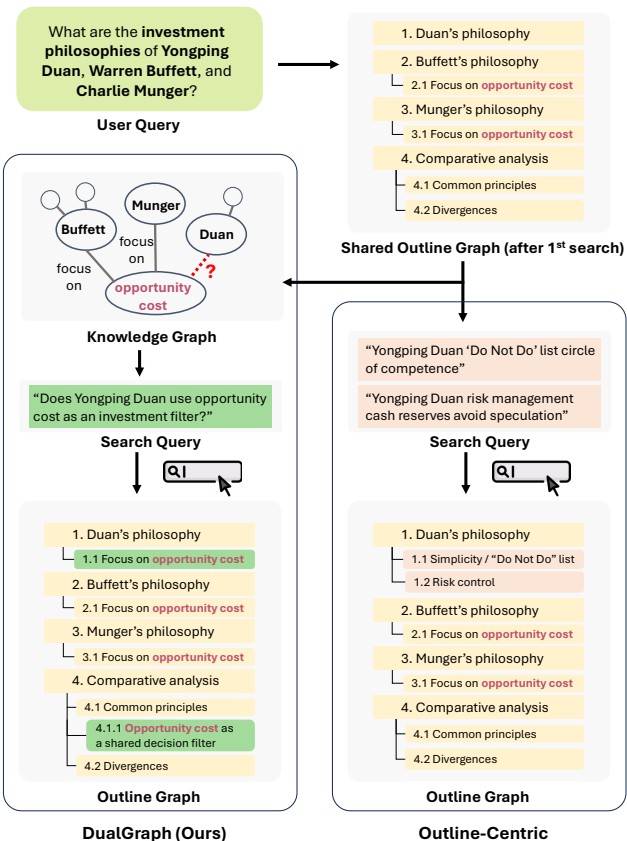

*Figure 1.* Motivating example contrasting DualGraph (left) and outline-centric (right) approaches. Starting from the shared outline graph, the outline-centric baseline continues to elaborate *Duan's philosophy* generally, fails to explore whether *Duan* emphasizes *opportunity cost*, resulting in an incomplete comparison. In contrast, DualGraph exposes the missing *Duan–opportunity cost* relation in the Knowledge Graph and triggers a targeted in-depth query for a more comprehensive analysis.

another representation of underlying knowledge beyond the outline representation that can be referenced across outline sections while remaining uniquely stored.

To address these bottlenecks, we introduce **DualGraph**, a dual-graph memory architecture that separates knowledge exploration from outline structure. As illustrated in Figure 1, our system maintains two co-evolving structures: (a) The Outline Graph (OG): A hierarchical tree representing the sparse presentation layer (sections and subsections), optimized for narrative flow and readability, representing *how* the deep research agent writes the final report. (b) The Knowledge Graph (KG): A semantic graph serving as the dense and unified representation of knowledge, storing atomic knowledge units (core entities, concepts) uniquely, preventing duplication even if referenced by multiple outline sections. It represents *what* the agent knows.

Crucially, the KG is not only a memory structure, but also an explicit epistemic state that enables the agent to reason

about knowledge coverage, gaps, disconnected claims, and missing relations by topological analysis. Together, the OG and KG form a joint exploration controller: the KG reveals missing or underexplored relations, while the OG provides structural context to prioritize and guide search, jointly generating targeted queries to resolve these gaps. Importantly, the OG and KG are co-evolved over iterative research rounds, with new evidence continuously reshaping both the report structure and the underlying epistemic state.

Empirically, we find that DualGraph leads to substantially more effective exploration and grounding across diverse research domains. On DeepResearch Bench (Du et al., 2025), DeepResearch Bench II (Li et al., 2026a), DeepResearch-Gym (Coelho et al., 2025), and DeepConsult (youdotcom-oss, 2025), DualGraph consistently outperforms state-of-the-art baselines in terms of report depth, breadth, and factual grounding, i.e., reaching 53.08 RACE on DeepResearch Bench with GPT-5, 41.48 Total on DeepResearch Bench II, and achieving 94.31 average on DeepResearchGym and 6.42 on DeepConsult with GPT-4.1, while converging in fewer research iterations. Ablation studies further show that the explicit use of KG to guide search leads to more efficient exploration and higher-quality research results, confirming that DualGraph is critical for effective long-horizon deep research. Our contributions are as follows:

- We propose the DualGraph memory framework, which separates knowledge representation from report structure, enabling agents to maintain a semantic knowledge graph distinct from the report outline.
- We introduce a graph-driven exploration mechanism, where search queries are derived from knowledge graph topology (e.g., missing links, under-supported nodes), allowing agents to systematically identify knowledge gaps and guide exploration.
- We demonstrate via experiments on four OEDR benchmarks that DualGraph leads to improved exploration in deep research iterations and the final reports quality.

## 2. Related Works

### 2.1. Open-Ended Deep Research Agent

A large line of web research agents (Huang et al., 2025) targets complex but relatively short question answering: they interleave search and navigation with reasoning and return a bounded answer, often optimized around QA-style benchmarks (Team et al., 2025; Zhang et al., 2026; Qiao et al., 2025; Li et al., 2025a; Tao et al., 2026; Li et al., 2025c; Wu et al., 2025). These systems emphasize finding the right evidence and producing a correct answer, rather than constructing a coherent multi-section report (Wei et al., 2025). In contrast, open-ended deep research (OEDR) asks for analyst-style, long-form, citation-

rich reports, evaluated by report quality and citation accuracy on benchmarks such as DeepResearch Bench (Du et al., 2025), DeepResearchGym (Coelho et al., 2025), and DeepConsult (youdotcom-oss, 2025). Recent proprietary products show strong OEDR capability (e.g., OpenAI Deep Research (OpenAI, 2025), Gemini Deep Research (Google, 2025), and Claude Research (Anthropic, 2025)). Meanwhile, open-source efforts span two paradigms: (i) "search-then-generate" pipelines (Han et al., 2025; LangChain; Felovic; Team et al., 2025) that gather many documents and draft a report in one pass, but this static structure can harm coherence and factual grounding (Li et al., 2026b). (ii) outline-centric systems (Shao et al., 2024) that draft an outline or framework and iteratively refine it with retrieved content before final report generation (Xiong et al., 2025; Li et al., 2026b). In contrast, our dual-graph approach co-evolves an outline graph and a knowledge graph, making it easy to identify knowledge gaps, missing bridges, and contradictions, and to turn them into targeted follow-up search queries.

## 2.2. Graph-Based Memory for Deep Research

Long-horizon agents depend on memory mechanisms to store, organize, and retrieve knowledge across extended trajectories, such as MemGPT (Packer et al., 2023), Mem0 (Chhikara et al., 2025), and A-MEM (Xu et al., 2025). In parallel, graph-augmented retrieval (e.g., GraphRAG-style methods (Edge et al., 2024; Jimenez Gutierrez et al., 2024; Zhu et al., 2025)) uses graphs to improve query-focused synthesis over corpora. However, much of this memory work is motivated by multi-session conversation, personalization, or general long-context limits. For OEDR, what's often missing is a memory design tailored to iterative knowledge acquisition, i.e., memory that does not merely "store evidence", but actively helps the agent decide what is missing, and what connections should be investigated next. Even in strong OEDR systems (Li et al., 2026b), memory is often a passive evidence store rather than an active representation that shapes exploration. Our dual-graph approach instead uses a knowledge-graph–centric memory to encode knowledge units and relations, so gap signals, e.g., weak support, missing links, contradictions, can directly drive follow-up queries and outline updates, making exploration knowledge-driven rather than outline-driven. Appendix A.4 further discusses the distinction from graph-based retrieval agents.

## 3. Methodology

### 3.1. Preliminary

**Outline Graph (OG).** OG captures the evolving report organization as a hierarchical tree of sections and subsections, where each node contains a brief semantic description (see

Figure 1 for examples). Unlike static planning, OG is iteratively refined as new evidence is retrieved, allowing sections to be expanded, restructured, or re-scoped. Each section node is associated with a set of grounded citations $C_i$ (evidence IDs) that link OG directly to entries in the evidence bank, where the collected evidence is stored.

**Knowledge Graph (KG).** We maintain a global, iteration-accumulated representation of knowledge as an edge-grounded graph $KG = (V, E)$. The node set $V$ includes (i) *Core-Entity* nodes for primary subjects in the user query (e.g., *Duan*, *Buffett*, *Munger*) as well as newly explored subjects, and (ii) *Concept* nodes for salient principles, attributes, or themes discussed across sources (e.g., *opportunity cost*). Each edge $e = (u, v, r; \{C_i\}) \in E$ links nodes $u$ and $v$ with a relation $r$ (directed when appropriate) and is grounded by a set of evidence IDs $C_i$ that point to the supporting evidence units in the evidence bank. This design stores shared knowledge once and makes it easier to reason about coverage and missing relations for downstream exploration.

**OEDR Workflow.** Open-Ended Deep Research (OEDR) takes a user query and produces a structured long-form report grounded in external evidence. We follow an iterative retrieval-and-writing workflow. Given the query, the agent first generates an initial OG (without citations), and uses this OG to propose a first round of search queries, which are issued to a search engine to retrieve relevant documents. The retrieved content is processed into an evidence bank of citable evidence units with unique IDs. After this initialization round, outline-centric methods update the OG using the collected evidence and generate the next-round queries from the updated OG alone. In contrast, DualGraph additionally constructs a KG from the evidence and co-updates both OG and KG, and subsequent search queries are inferred by jointly leveraging OG for structural context and KG for reasoning about coverage and missing relations. After several iterations, the agent writes the report in a section-by-section manner by retrieving the cited evidence units from the evidence bank and synthesizing each section from them.

### 3.2. Overview of DualGraph

Figure 2 provides an overview of DualGraph. ❶ Given a user query, the agent initializes an OG *w/o* citations using internal knowledge, derives an initial set of search queries, and issues them to a search engine. Then the retrieved documents are processed into an evidence bank of citable evidence units. ❷ Using this initial evidence, the agent updates the OG by refining the outline and attaching grounded citations (evidence IDs). In parallel, ❸ it extracts atomic knowledge units and relations from the evidence to construct an initial KG or update KG in later iterations. ❹ DualGraph then performs a KG-driven gap discovery, identifying weakly supported or missing relations to form search

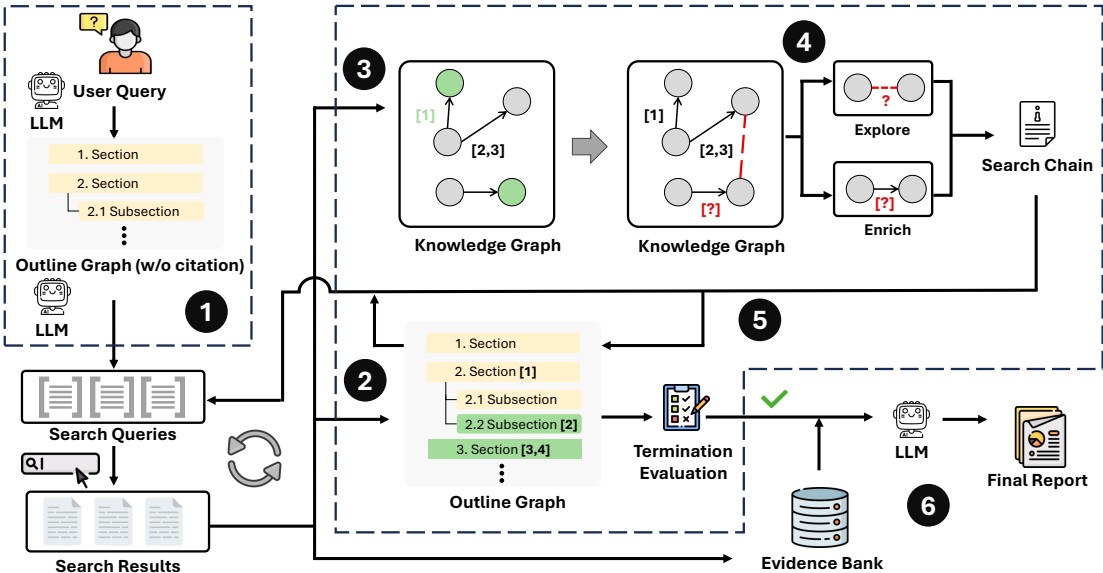

*Figure 2.* Overview of our DualGraph framework. ❶ shows the OG initialization, ❷-❺ shows the co-evolving iteration of KG and OG, and ❻ shows the report finalization after the iteration termination.

chains for *enrichment* and *exploration*. ❺ These search chains, together with the OG's structural context, are converted into the next-round search queries. The newly retrieved evidence is used to co-update both OG and KG, repeating the iteration until an early-termination criterion is met or a maximum number of rounds is reached. ❻ DualGraph writes the report section by section following the finalized OG, retrieving only the evidence units referenced by each section's citations from the evidence bank. Algorithmic details are given in Appendix B.3, and Appendix H provides a qualitative case trajectory illustrating the iterative workflow. Similar to the OEDR workflow in Section 3.1, ❶ initializes the OG following traditional outline-centric methods. In the following sections, we detail the KG–OG co-evolving iterations (❷–❺) and the section-wise report finalization stage (❻).

## 3.3. Co-Evolution of Outline and Knowledge Graph

Unlike outline-centric methods, DualGraph maintains both an OG and a KG throughout the OEDR workflow. OG captures the evolving report organization (i.e., how the agent structures and writes the report), while KG maintains the underlying knowledge state (i.e., what the agent has learned so far and how facts are connected). KG-driven search chain generation explicitly identifies missing or weakly supported relations that should be explored or enriched. The OG provides complementary structural context to broaden coverage in the KG-driven search chains. The resulting search queries retrieve new evidence, which is then used to jointly update both KG and OG, forming a closed-loop co-evolution process.

❷ **Outline Graph Update and Citation Maintenance.** Given new search results, we extract a concise evidence unit from each result using an LLM. Rather than summarizing the entire webpage, we condition extraction on the originating search query to retain only query-relevant evidence (see the pipeline in Appendix B.1). The newly acquired evidence is then used to revise the current OG in two ways. First, the agent updates the outline structure by adding new sections/subsections, revising existing ones, and removing redundant items, since newly discovered evidence may introduce missing aspects or correct unsupported outline content. Second, the agent grounds OG sections with citation sets $C_i$ by attaching evidence IDs that support each section.

Crucially, DualGraph enforces citation persistence during this process. When sections are restructured, expanded, or modified, their existing citations are preserved and consistently propagated. This constraint keeps outline evolution synchronized with accumulated evidence, preventing citation loss and maintaining traceability across iterations.

❸ **Knowledge Graph Update and Citation Maintenance.** On the knowledge side, DualGraph expands and refines KG by incorporating newly extracted evidence from search results. Given a batch of extracted evidence units, KG is updated in four steps. (1) Knowledge node and relation extraction: conditioned on the root user query and the issued search queries, we use an LLM to extract entities, concepts, and relations, and insert the corresponding nodes and edges into KG (see the prompt in Appendix I.5). The citations are associated with the newly added or updated edges in KG. (2) Semantic merging and canonicalization: semantically equivalent or highly overlapping nodes are merged into a

canonical node, with edges and evidence IDs redirected accordingly. (3) Semantic clustering: nodes are grouped into semantically coherent clusters based on embedding similarity and annotated with cluster IDs to provide higher-level semantic context. (4) Community detection: to capture topological groupings beyond embedding similarity, we apply Leiden community detection (Traag et al., 2019; Edge et al., 2024) to partition the KG into densely connected communities, where nodes are assigned community IDs that reflect the evolving topology.

❹ **KG-Driven Knowledge Gap Discovery.** At each iteration, DualGraph analyzes the current KG to propose targeted search queries. Specifically, it inspects the KG's semantic and topological structure together with evidence coverage, and constructs a set of search chains that serve as intermediate cues for search query generation. Each search chain serves as a structured candidate for query generation and falls into one of two categories; the LLM then filters and selects appropriate chains (see Appendix C).

*Evidence Enrichment (Enrich).* For relations already present in the KG, DualGraph selects edges with weak evidence support as enrichment targets. Candidates are ranked using both node-level importance (e.g., whether the edge involves core entities) and expected structural impact (e.g., edges that connect different communities are prioritized), so the agent can strengthen evidence support for critical relations with minimal additional retrieval (see Appendix C.1 for details). This enrichment improves traceability and reduces the risk of weak grounding during report writing.

*Knowledge Discovery (Explore).* For relations that are not yet explicitly established in the current KG but are potentially important for completing the overall research landscape, DualGraph constructs exploratory chains to drive knowledge gap discovery. It combines multiple complementary and graph-internal signals to identify promising relations (see Appendix C.2 for details), including: (1) semantic similarity for candidate links between core entities and concept nodes; (2) community-level global gap statistics, where a stochastic block model (SBM) (Abbe, 2018) is used to estimate the prior probability and uncertainty of cross-community links; (3) cross-community bridging and structural-hole cues (Burt, 2004) to uncover connections that may facilitate cross-topic knowledge integration. By jointly considering these signals, these signals balance high-confidence completion with high-value exploration, helping the agent systematically reduce structural gaps in the KG.

❺ **Search Query Generation.** Given the OG and the search chains derived from the updated KG, we use an LLM to generate concrete search queries (see the prompt in Appendix I.3). In addition, we generate supplementary queries from the updated OG. Although many OG sections may still lack grounded citations, they capture planned aspects that remain under-explored and are therefore valuable for retrieval. Since KG-driven search chains are largely related to already grounded nodes and relations (extracted from existing evidence), they can miss parts of the query space that have not yet been incorporated into KG. We therefore provide both the KG-derived search chains and the updated OG to LLM to produce a merged set of queries with broader coverage.

### 3.4. Report Generation

**Self-Termination with OG.** The co-evolving loop continues until a termination evaluator judges that further retrieval is unlikely to add meaningful information. Following (Li et al., 2026b), it scores the current OG along several dimensions (see the Appendix B.2): *instruction following* (how well the outline matches the user request), *depth* (level of detail and analysis), *breadth* (coverage of distinct relevant aspects), *balance* (fairness and objectivity), *support* (quality of the evidence plan and citations), and *insightfulness* (usefulness and non-obvious takeaways). The iteration stops when all scores exceed predefined thresholds, avoiding redundant searching.

❻ **Report Writing.** Once self-terminated, the writer generates the report section by section. For each OG section, the system uses its grounded citation set $C_i$ to retrieve the corresponding evidence units from the evidence bank, and synthesizes the section content.

## 4. Experiments

### 4.1. Experimental Setup

**Benchmarks.** To comprehensively assess the capabilities of Deep Research systems in addressing OEDR problems, we adopt four publicly available benchmark datasets. More details are provided in Appendix A.

*DeepResearch Bench* (Du et al., 2025) contains 100 highly challenging research tasks designed at the PhD level. These tasks are curated by subject-matter experts and span 22 diverse domains, including science and technology, finance and business, software engineering, and art and design.

*DeepResearch Bench II* (Li et al., 2026a) improves upon the original benchmark in two key aspects: (1) it replaces the FACT metric with InfoRecall for evaluating information retrieval capability, providing a more direct measure of evidence coverage; and (2) it adopts fine-grained, fully verifiable rubrics that do not rely on the judge model's internal domain knowledge, ensuring more objective evaluation.

*DeepConsult* (youdotcom-oss, 2025) focuses on in-depth research scenarios in business and consulting contexts. It consists of domain-specific queries covering a broad spectrum of topics such as market analysis, financial modeling,

strategic planning, and emerging technological trends.

*DeepResearchGym* (Coelho et al., 2025) is employed to evaluate system performance on realistic and complex information-seeking tasks. It includes 100 queries sampled from the Researchy Questions dataset (Rosset et al., 2024), which comprises approximately 96,000 real-world research questions collected from diverse sources.

**Metrics.** For each benchmark, we follow the official evaluation protocols and employ the recommended LLMs as judges (Liu et al., 2024) to evaluate system performance. A human evaluation study validating the reliability of LLM-as-a-Judge is provided in Appendix G.

*DeepResearch Bench.* This benchmark evaluates system outputs using two complementary metric groups that target different dimensions of research quality: (1) *RACE (Report Quality)* measures the overall quality of the generated report by comparing it against a reference report along four aspects: comprehensiveness (Comp.), depth and insight (Insight), adherence to instructions (Inst.), and readability (Read.). The final RACE score is computed as a weighted aggregation of these components. (2) *FACT (Web Retrieval)* focuses on the effectiveness and reliability of the information retrieval process, including citation accuracy (C.acc.) and the average number of effective citations per task (Eff.c.).

*DeepResearch Bench II.* This benchmark evaluates reports along three dimensions: InfoRecall (whether the report accurately and comprehensively retrieves relevant facts and evidence), Analysis (whether the report forms higher-level conclusions from retrieved evidence), and Presentation (overall writing quality).

*DeepConsult.* Evaluation on DeepConsult is conducted through pairwise comparisons against the reports generated by *openai-deepsearch* (OpenAI, 2025) baseline. Model performance is primarily quantified using win, tie, and loss rates, which are further complemented by a reported average quality score.

*DeepResearchGym.* For DeepResearchGym, an LLM judge assesses each generated report across multiple qualitative dimensions, including clarity, insightfulness, depth, balance, breadth, and evidential support. An overall average quality score is computed based on these evaluations.

**Baselines.** We assess the performance of DualGraph by comparing it with a diverse set of representative Deep Research systems. Following common practice, the methods are grouped into open-source and proprietary systems.

*Open-Source Systems.* For open-source baselines, we consider `LangChain-DeepResearch` (LangChain) as a modular orchestration framework for deep research pipelines, as well as recent open-weight LLMs from the Qwen (QwenLM, 2026) and Tongyi (Team et al., 2025) families that support multi-step reasoning and tool-augmented research workflows. We also include WebShaper (Tao et al., 2026), which formalizes information-seeking tasks via set-theoretic knowledge projections to synthesize high-quality training data for web-based agents, and WebWeaver (Li et al., 2026b), an outline-centric deep research that iteratively refines a dynamic outline while acquiring evidence.

*Proprietary Systems.* We evaluate our method against several state-of-the-art commercial Deep Research systems, including `Claude-research` (Anthropic, 2025), `OpenAI DeepResearch` (OpenAI, 2025), and `Gemini-2.5-pro-deepresearch` (Google, 2025).

**Implementation Details.** We conduct experiments using two backend LLMs to ensure robustness across different model configurations, namely `gpt-4.1-20250414` and `gpt-5-chat-20250807`. Web search is performed via the `Bing Search API v7` (Microsoft, 2026), and URL visiting and page parsing are handled using the `Crawl4AI` (UncleCode, 2024). We set the maximum number of optimization rounds to `MAX_ITER=5`. More implementation details are provided in Appendix B.

### 4.2. Main Results

**Results on DeepResearch Bench.** To align the LLM judge versions, we obtain the reports of proprietary systems from DeepResearch Bench and re-evaluate them using the same judge model. As shown in Table 1, DualGraph is consistently strong across backbones and competitive with leading proprietary systems: (1) With GPT-5, DualGraph achieves the best overall RACE of 53.08, comparable to Gemini-2.5-Pro-DeepResearch at 52.54. (2) Scaling the backbone from GPT-4.1 to GPT-5 yields consistent gains: RACE increases from 51.50 to 53.08 (+1.58) for DualGraph, and from 49.18 to 51.17 (+1.99) for the w/o KG variant. (3) Adding KG improves report quality beyond overall: with GPT-5 it raises comprehensiveness from 51.69 to 53.78 (+2.09) and Insight from 51.79 to 54.30 (+2.51), while instruction-following or readability change modestly, suggesting KG mainly helps surface missing aspects and deepen analysis rather than formatting. (4) KG also substantially increases evidence coverage: with GPT-5, effective citations rises from 66.06 to 79.65 (+13.59), indicating more effective incorporation of supporting evidence via gap-driven exploration.

Introducing KG slightly reduces citation accuracy (C.acc.) but substantially increases effective citations (Eff. C.). Concretely, with GPT-5, C.acc. drops from 61.15 to 57.55 (−3.60) while Eff. C. rises from 66.06 to 79.65 (+13.59); with GPT-4.1, C.acc. decreases from 59.96 to 56.23 (−3.73) but Eff. C. increases from 62.96 to 76.33 (+13.37). We

*Table 1.* DeepResearch Bench results across RACE and FACT. Higher is better. Missing results are denoted by "–". Abbreviations: DR = DeepResearch; Res. = Research; RA = Research Assistant; KG = Knowledge Graph; Comp. = Comprehensiveness; Inst. = Instruction-following; Read. = Readability; Eff.c. = Effective citations; C.acc. = Citation accuracy. Note that LangChain-DeepResearch / Qwen-Agent / Tongyi-DeepResearch / Web-Weaver results are using GPT-4.1 as the backbone LLM unless otherwise specified. WebShaper uses the Alibaba-NLP/WebShaper-32B model.

| Agent Systems | RACE | | | | | FACT | |
|---|---|---|---|---|---|---|---|
| | Overall | Comp. | Insight | Inst. | Read. | C. acc. | Eff. c. |
| Claude-3.5 (Search) | 34.32 | 33.43 | 32.37 | 36.98 | 36.34 | 93.80 | 8.96 |
| Claude-3.7 (Search) | 41.56 | 41.42 | 39.99 | 43.80 | 42.02 | 87.32 | 24.51 |
| Doubao-DR | 46.62 | 47.67 | 45.21 | 48.16 | 45.07 | 51.87 | 35.12 |
| Gemini-2.5-Pro (DR) | 52.54 | 52.65 | 53.20 | 51.82 | 51.83 | 72.11 | 53.38 |
| GPT-4.1 | 43.08 | 42.85 | 41.96 | 44.62 | 43.77 | 85.75 | 3.84 |
| GPT-4.1-mini | 34.21 | 33.36 | 30.52 | 38.18 | 37.64 | 82.63 | 4.08 |
| Grok (Deep Search) | 41.01 | 40.55 | 38.82 | 43.56 | 42.83 | 68.47 | 7.28 |
| Kimi (Res.) | 46.50 | 47.06 | 45.67 | 47.53 | 45.47 | 20.80 | 2.30 |
| NVIDIA AIQ (RA) | 42.48 | 42.01 | 41.81 | 43.80 | 43.01 | 46.01 | 4.41 |
| OpenAI-DR | 46.68 | 47.27 | 45.55 | 48.00 | 45.93 | 73.18 | 31.70 |
| Perplexity (Res.) | 42.44 | 42.36 | 40.85 | 44.41 | 43.11 | 76.03 | 21.69 |
| Sonar | 37.29 | 36.26 | 34.51 | 40.46 | 40.61 | 69.59 | 8.11 |
| Sonar-Pro | 40.31 | 39.76 | 38.35 | 42.68 | 42.02 | 76.64 | 13.06 |
| Sonar-Reason | 40.53 | 39.73 | 38.87 | 42.79 | 42.26 | 51.19 | 11.60 |
| Sonar-Reason-Pro | 40.74 | 40.00 | 39.06 | 43.14 | 42.30 | 37.31 | 6.72 |
| LangChain-DR | 42.92 | 41.95 | 44.63 | 43.75 | 43.08 | 59.83 | 61.29 |
| Qwen-Agent | 40.95 | 39.19 | 43.65 | 42.88 | 41.27 | 52.23 | 8.39 |
| Tongyi-DR | 40.62 | 38.63 | 43.92 | 42.58 | 41.03 | – | – |
| WebWeaver | 45.20 | 45.53 | 44.08 | 46.39 | 45.36 | 40.50 | 48.59 |
| WebShaper | 40.28 | 39.58 | 38.21 | 42.71 | 42.42 | 39.23 | 4.43 |
| DualGraph w/o KG (GPT-4.1) | 49.18 | 50.13 | 48.41 | 50.06 | 47.35 | 59.96 | 62.96 |
| DualGraph w/o KG (GPT-5) | 51.17 | 51.69 | 51.79 | 51.16 | 48.75 | 61.15 | 66.06 |
| **DualGraph (GPT-4.1)** | 51.50 | 52.07 | 52.00 | 51.41 | 49.30 | 56.23 | 76.33 |
| **DualGraph (GPT-5)** | 53.08 | 53.78 | 54.30 | 52.48 | 49.81 | 57.55 | 79.65 |

attribute this C.acc. pattern primarily to the per-citation evaluation protocol of FACT: C.acc checks whether each citation *independently* supports its claim, which systematically underestimates accuracy when a claim is synthesized from multiple sources (see Appendix A.2 for a detailed analysis and joint-evaluation experiment confirming this).

**Results on DeepResearchGym and DeepConsult.** Table 2 further demonstrates the versatility and robustness of DualGraph across diverse research scenarios. On *DeepResearchGym*, DualGraph (GPT-4.1) achieves the best overall performance among all open-source systems and the majority of proprietary systems, reaching an Avg. score of 94.31. It exhibits near-saturated performance on coverage-related criteria (e.g., Depth and Breadth), while also maintaining strong results on evidence- and insight-oriented dimensions. Compared with the w/o-KG variant, introducing the iteratively constructed KG leads to consistent improvements across all reported metrics.

On the *DeepConsult* benchmark, DualGraph (*GPT-4.1*) further demonstrates strong practical effectiveness, achieving a win rate of 64.42% and outperforming its w/o-KG counterpart. These results indicate that explicit knowledge structuring helps the agent maintain comprehensive coverage and

produce better-supported, higher-quality syntheses.

**Results on DeepResearch Bench II.** As shown in Table 3, DualGraph achieves a Total score of 41.48. Moreover, DualGraph consistently outperforms DualGraph w/o KG across all metrics, particularly on InfoRecall (34.21 vs. 32.18) and Analysis (52.45 vs. 50.73), confirming that the KG component strengthens both information retrieval and analytical reasoning. This improvement on InfoRecall supports our analysis in Appendix A.2: the more targeted InfoRecall metric reveals that KG-driven exploration genuinely enhances the breadth and relevance of retrieved information.

### 4.3. Ablation Study

**Analysis of iteration rounds.** On the DeepResearch Bench, we report the number of refinement iterations until termination. Table 4 indicates that DualGraph terminates in fewer refinement rounds than its counterpart without the knowledge graph (KG). Specifically, DualGraph achieves a lower mean termination iteration (3.23 vs. 3.88) and a substantially higher fraction of early stops within two rounds, suggesting faster convergence. These results imply that KG grounding improves the efficiency of the iterative search process by guiding exploration toward more promising revisions, thereby reducing redundant planning steps and the overall number of refinement cycles required to reach a satisfactory solution.

**Outline quality across optimization rounds.** On the DeepResearch Bench, to more directly quantify the contribution of the KG during iterative planning, we examine intermediate outlines produced over five optimization iterations. Following the same LLM-as-a-judge protocol, we score each outline along four axes which are directly relevant to the knowledge quality and coverage: *Insightfulness*, *Depth*, *Breadth*, and *Support*, where higher scores indicate better quality. As shown in Figure 4, both variants generally improve with additional iterations, confirming the general effectiveness of iterative refinement. Notably, the KG-enabled variant exhibits consistently steeper gains and reaches higher final performance across all dimensions. The largest margin appears in *Support*: KG guidance rapidly increases evidence- and citation-grounded content (reaching the mid-60s by Iter 3–4), whereas the outline-only baseline improves more slowly and remains in the low-to-mid 50s even by Iter 5. Persistent advantages are also observed in *Breadth* and *Depth*, where the KG variant approaches near-saturated breadth (approximately 99 by Iter 5) and stronger depth (approximately 90), while the baseline lags by several points. These trends suggest that KG guidance helps the planner surface missing relations and diversify subtopics earlier in the optimization trajectory, accelerating convergence to a high-quality research blueprint.

*Table 2.* Performance on DeepConsult (win rate and average score) and DeepResearchGym (including Avg.). Higher is better. Column meanings: Cla. = Clarity; Bal. = Balance; Brea. = Breadth; Sup. = Support; Ins. = Insightfulness. Notes: LangChain-DeepResearch / Qwen-Agent / Tongyi-DeepResearch / WebWeaver results use GPT-4.1. WebShaper uses the Alibaba-NLP/WebShaper-32B model. Results marked with * are taken from Li et al. (2026b).

| Agent Systems | DeepResearchGym | | | | | | | DeepConsult | | | |
|---|---|---|---|---|---|---|---|---|---|---|---|
| | Cla. | Depth | Bal. | Brea. | Sup. | Ins. | Avg. | Win | Tie | Lose | Avg. Score |
| Claude-Research* | 86.67 | 96.88 | 84.41 | 96.56 | 26.77 | 90.22 | 80.25 | 25.00 | 38.89 | 36.11 | 4.60 |
| OpenAI-DeepResearch* | 84.90 | 98.10 | 89.80 | 97.40 | 88.40 | 89.00 | 91.27 | 0.00 | 100.00 | 0.00 | 5.00 |
| Doubao-Research* | 68.85 | 93.12 | 83.96 | 93.33 | 84.38 | 83.12 | 84.46 | 29.95 | 40.35 | 29.70 | 5.42 |
| Gemini-2.5-Pro-DeepResearch* | 90.71 | 99.90 | 93.37 | 99.69 | 95.00 | 97.45 | 96.02 | 61.27 | 31.13 | 7.60 | 6.70 |
| LangChain-DeepResearch | 85.20 | 93.30 | 90.50 | 96.90 | 85.90 | 85.90 | 89.62 | 14.46 | 44.60 | 40.90 | 4.26 |
| Qwen-Agent | 64.70 | 82.40 | 82.70 | 81.00 | 64.70 | 60.40 | 72.65 | 4.41 | 4.41 | 91.17 | 1.99 |
| Tongyi-DeepResearch | 77.50 | 72.70 | 83.10 | 86.40 | 16.80 | 62.50 | 66.50 | 13.48 | 10.54 | 75.98 | 2.63 |
| WebShaper | 80.10 | 85.90 | 81.80 | 86.40 | 71.00 | 75.80 | 80.17 | 15.93 | 19.61 | 64.46 | 3.71 |
| WebWeaver | 77.20 | 99.30 | 88.70 | 99.30 | 89.70 | 90.80 | 90.84 | 50.77 | 22.70 | 26.53 | 5.65 |
| DualGraph w/o KG (GPT-4.1) | 87.11 | 99.95 | 92.16 | 99.95 | 89.66 | 93.24 | 93.68 | 57.61 | 28.26 | 14.13 | 6.24 |
| DualGraph (GPT-4.1) | 88.14 | 99.99 | 92.37 | 99.98 | 91.86 | 93.52 | 94.31 | 64.42 | 23.08 | 12.50 | 6.42 |
| DualGraph (GPT-5) | 90.73 | 99.98 | 93.80 | 99.96 | 97.20 | 97.48 | 96.52 | 67.27 | 16.24 | 16.49 | 6.84 |

*Table 3.* Performance comparison on DeepResearch Bench II. Higher is better.

| Agent System | Total | InfoRecall | Analysis | Presentation |
|---|---|---|---|---|
| Gemini2.5-Pro-DR | 45.89 | 38.66 | 59.08 | 92.10 |
| Perplexity-Research | 39.60 | 33.06 | 49.62 | 83.09 |
| Grok-DeepSearch | 38.85 | 32.08 | 47.93 | 90.51 |
| WebWeaver | 36.57 | 29.58 | 46.81 | 86.57 |
| WebShaper | 27.88 | 21.80 | 28.68 | 87.05 |
| DualGraph w/o KG (GPT-4.1) | 39.50 | 32.18 | 50.73 | 90.27 |
| **DualGraph (GPT-4.1)** | 41.48 | 34.21 | 52.45 | 90.65 |

*Table 4.* Distribution of termination iterations across methods on DeepResearch Bench. Lower Avg. Iter indicates faster convergence.

| Method | Iter 1 | Iter 2 | Iter 3 | Iter 4 | Iter 5 | Avg. Iter |
|---|---|---|---|---|---|---|
| DualGraph | 1% | 34% | 25% | 21% | 19% | 3.23 |
| DualGraph w/o KG | 2% | 11% | 25% | 21% | 41% | 3.88 |

We further substantiate this finding by directly using an LLM as a judge to compare the quality of intermediate outlines produced by the two methods on DeepResearch Bench. Specifically, we conduct pairwise win-rate evaluations between KG-enhanced and OG-only outlines across iterations, as shown in Figure 5. The KG-enhanced outlines win 53.3% of cases at Iter 1, rising to 63.0% (Iter 2) and 73.3% (Iter 3), and continuing to improve to 75.1% (Iter 4) and 76.2% (Iter 5). The sharp gains around Iter 2–3 align with the plateauing behavior in the score curves, suggesting that KG guidance reaches high-quality outlines in fewer iterations and thereby motivates an early-stopping strategy to reduce computation cost without sacrificing outline quality.

**Fine-grained ablation study.** On the DeepResearch Bench, to isolate the contribution of each component, we compare DualGraph against five variants with OG-based self-termination disabled for all: (1) **w/o KG** removes the entire KG; (2) **w/o Enrich** disables enrichment search chains; (3) **w/o Explore** disables exploratory search chains; (4) **only EntityRelation** just tracking entity-relation gaps; (5) **by LightRAG** replaces our KG construction with LightRAG (Guo et al., 2025)'s retrieval-oriented implementation. As shown in Figure 3, DualGraph consistently outperforms all variants across iterations on both outline quality (left) and cumulative search query and evidence quality (right); more details see Appendix A.3. Removing either Enrich or Explore chains leads to degradation, confirming both chain types are indispensable. The only EntityRelation variant also underperforms, confirming that topological signals beyond raw entity-relation gaps (community structure, SBM, structural holes) are essential. The LightRAG variant performs significantly worse, suggesting that retrieval-oriented KG construction (producing overly fine-grained triplets) is not suited for gap detection.

## 5. Discussion and Limitations

A primary limitation of DualGraph is the additional computation and token overhead introduced by explicitly constructing and maintaining the KG during iterative research. Compared to the outline-only variant (DualGraph w/o KG), KG updates require extra LLM calls for knowledge extraction, relation refinement, and graph consolidation. However, our cost analysis in Appendix E indicates that this overhead is modest relative to the dominant Web Search/Crawl pipeline (LLM-based page reading and summarization), which accounts for the majority of end-to-end tokens. In particular,

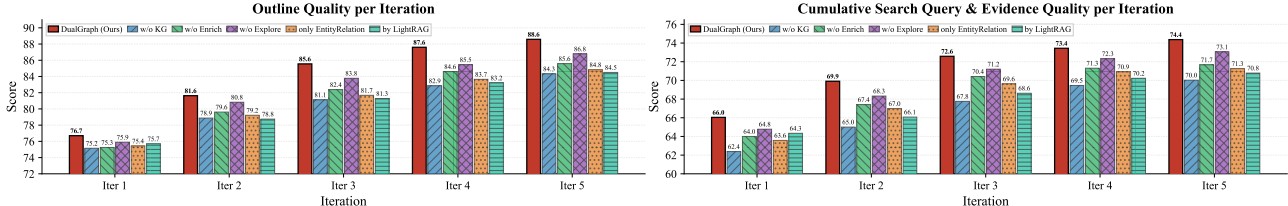

*Figure 3.* Comparison of DualGraph variants across iterations. Early stopping is disabled for all variants to ensure fair comparison. Left: Outline quality. Right: Search query and evidence quality.

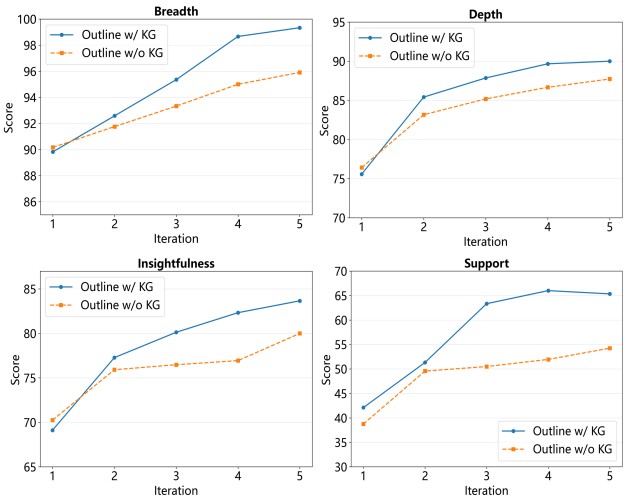

*Figure 4.* LLM-as-a-judge scores of intermediate outlines generated by DualGraph and DualGraph w/o KG across optimization iterations on DeepResearch Bench.

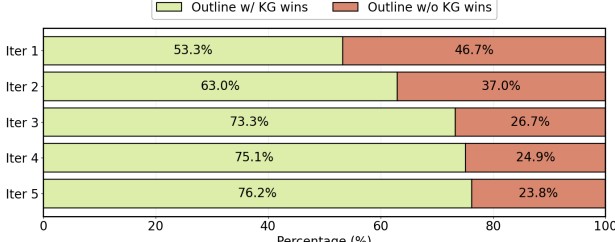

*Figure 5.* LLM-as-a-judge comparison of intermediate outlines generated by DualGraph and DualGraph w/o KG across optimization iterations on DeepResearch Bench.

the additional tokens attributable to KG maintenance are a small fraction of the total budget, and the early-stopping mechanism further mitigates the impact by reducing unnecessary refinement rounds.

Additionally, the quality of the constructed KG depends on the underlying LLM's ability to accurately abstract entities and relations from retrieved evidence. Imperfect or biased extractions may propagate into subsequent planning steps and influence exploration trajectories. However, our empirical analysis in Appendix F shows that the anomaly rate is low (2.4%), and only 0.12% of all triplets are both anomalous and used before self-correction, thanks to the iterative merge-and-update mechanism built into DualGraph. Finally, our work focuses on text-centric deep research benchmarks, and the effectiveness of DualGraph design in multimodal research settings remains to be explored.

## 6. Conclusion

In this work, we introduced DualGraph, a Dual-Graph memory architecture for open-ended deep research that separates knowledge representation from report structure, by maintaining a Knowledge Graph and jointly evolving it with a structured Outline Graph. Empirically, we showed that DualGraph leads to more systematic exploration and better-grounded research outputs across multiple challenging benchmarks. Ablation studies further show that our use of KG to guide the search leads to more efficient exploration and higher-quality research reports. More broadly, our results suggest that treating memory not as a passive storage but as an active guide for planning is a promising avenue for long-horizon reasoning agents.

## Impact Statement

This work aims to advance machine learning methods for improving long-horizon reasoning and structured decision-making in LLM agents. By enabling more systematic and efficient use of memory, the proposed approach has the potential to improve the reliability and usability of AI agents in productivity-oriented applications, as well as supporting knowledge discovery, scientific analysis, and other research-intensive tasks, which may yield positive societal benefits. We do not anticipate any immediate or unique societal harms beyond those commonly associated with LLM deployment.

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

# A. Eval Details

## A.1. Controlling the Effect of the Initial Outline and Initial Search

We observe that the initial outline can substantially influence the subsequent iterative trajectory and, ultimately, the final report quality. To ensure a fair ablation comparison, for the same root query we enforce that all ablation variants share exactly the same initialization at Iter $= 0$, including the outline, the set of search queries, and the corresponding retrieved search results. This control eliminates confounding factors introduced by different starting plans, so that performance differences can be attributed to the ablated components rather than initialization effects.

## A.2. About the FACT Metric in DeepResearch Bench

A low FACT Eff.c score for some baselines primarily reflects design choices rather than insufficient capability. Many systems intentionally prioritize source quality and non-redundancy over maximizing the number of citations: during retrieval and filtering, they may discard duplicated, weakly relevant, or low-credibility materials (e.g., Baidu Baijiahao posts that often reproduce content from higher-quality platforms such as Zhihu). Including such duplicated/low-quality sources can inflate FACT by increasing citation count or coverage, but it may contribute little to the report's actual informativeness or reliability. In addition, several baselines follow a ReAct-style paradigm in which the amount of retrieved evidence is largely determined by the base model: it stops searching once it deems the accumulated evidence sufficient, without an explicit externally enforced stopping rule. Consequently, a low FACT Eff.c does not necessarily imply poor overall quality; rather, it more directly indicates a weaker emphasis on citation intensity and redundancy. For example, Claude-3.5-Sonnet-with-Search often produces reports with only a small number of citations; if a report contains 8 citations in total, Eff.c is inherently upper-bounded by 8 and cannot exceed this limit. Therefore, FACT should not be treated as a standalone indicator of report quality, especially when higher scores may partly arise from redundant or low-value citations.

We also note that C.acc evaluates each citation independently, checking whether it alone can support the corresponding claim. This per-citation protocol is reasonable when claims are grounded in a single source, but may systematically underestimate accuracy when a claim is synthesized from multiple citations, as no single citation fully covers the joint conclusion. To examine whether this evaluation protocol accounts for the observed C.acc difference, we randomly sampled 20 cases and conducted a joint evaluation where all citations for a given claim are assessed together. Under this setting, DualGraph and the w/o KG variant achieve nearly identical accuracy (90.51% vs. 90.41%), suggesting that the C.acc difference largely stems from the per-citation evaluation protocol rather than lower citation quality. The recently released DeepResearch Bench II (Li et al., 2026a) addresses this limitation by replacing FACT with InfoRecall, a metric that directly measures whether the report accurately and comprehensively retrieves relevant facts and evidence. As shown in Table 3, DualGraph improves over the w/o KG variant on InfoRecall (34.21 vs. 32.18), further supporting that KG-driven exploration enhances evidence coverage without compromising accuracy.

## A.3. Details for Fine-grained Ablation

To conduct a more fine-grained ablation, we decompose the system beyond the presence or absence of the KG and further examine specific KG-driven mechanisms, such as different types of search chains and the way the knowledge graph is constructed. Accordingly, we move beyond evaluating only the final report and instead assess the intermediate products produced during the iterative process: the evolving outline and the accumulated search queries with retrieved evidence. This is necessary because the ablated components primarily affect the iterative process, rather than only the final writing stage.

In this fine-grained ablation study, we disable the OG-based self-termination mechanism and force all methods to run until Iter $= 5$. This provides a complete and comparable trajectory for all variants, preventing differences in stopping time from confounding the analysis. It also allows us to consistently evaluate the intermediate products produced throughout the full iterative process.

For outline quality, we follow the same LLM-as-a-judge protocol described in Paragraph 4.3, scoring intermediate outlines along the dimensions directly related to planning quality and evidence grounding. For search quality, we evaluate the accumulated retrieval products produced up to each iteration. Specifically, at iteration $t$, the judge is given all search queries and retrieved evidence summaries accumulated from iteration $0$ through iteration $t$, rather than only the newly generated queries and evidence in the current iteration. We ask the judge model to assess their combined quality under the DeepResearch Bench rubrics, focusing on *comprehensiveness* and *insight*: whether the accumulated queries target information needed for a high-quality final report, and whether the retrieved evidence is relevant, informative, and useful

for satisfying the corresponding rubric. The resulting score therefore reflects the overall quality of the accumulated search trajectory, rather than the isolated quality of a single iteration.

### A.4. Comparison with Graph-based Retrieval Agents

Recent graph-based retrieval and agent systems, such as LightRAG (Guo et al., 2025), Graph-R1 (Luo et al., 2025), GraphSearch (Yang et al., 2025a), and GraphAgent (Yang et al., 2025b), also use graph structures, but their role differs from the KG in DualGraph. LightRAG, Graph-R1, and GraphSearch primarily treat the graph as a retrieval index over a closed or preprocessed document corpus, where the graph stores existing knowledge and is queried to retrieve facts or passages for answering a given question. GraphAgent similarly operates over a fixed document graph as a reasoning substrate for downstream classification, summarization, or generation tasks.

In contrast, DualGraph targets open-ended long-form research without an available corpus in advance. The KG starts empty and is constructed incrementally from evidence retrieved during the iterative web research process. It is not used as a direct retrieval backend. Instead, DualGraph analyzes the evolving KG topology, including weakly supported edges, missing relations, community structure, and structural holes, to identify what knowledge is still missing and to generate follow-up search directions. This makes the KG a planning and gap-detection mechanism rather than a conventional graph retrieval index.

This difference also limits direct substitution of existing graph-retrieval pipelines. For this reason, our controlled comparison uses the LightRAG variant by replacing only the KG construction module while keeping the rest of DualGraph unchanged. As shown in Figure 3, this retrieval-oriented KG construction performs worse than the full design, supporting the need for a gap-detection-oriented KG in OEDR.

## B. Implementation Details

**Judging models.** Following the official benchmark guidelines, we adopt `gpt-4.1-20250414` as the judging model for DeepResearch Bench and DeepConsult, and `gpt-4.1-mini-20250414` for DeepResearchGym.

### B.1. Search and Evidence Acquisition Pipeline

The system maintains a global URL deduplication set to avoid repeated crawling across different queries and iterations. We adopt the below filtering pipeline for web retrieval and evidence acquisition:

**Web Search.** For each search query, we retrieve the top-$N$ candidate webpages from Bing.

**First-Stage Filtering (URL-level).** An LLM selects a subset of the candidate URLs based on their titles, snippets, and relevance to the query.

**URL Visit.** The selected URLs are visited and their webpage contents are parsed.

**Second-Stage Filtering (Evidence-level).** An LLM extracts structured evidence from the webpage content and determines whether it is useful. Only webpages judged as useful are retained as valid evidence. For each retained page, we preserve both salient content spans from the original webpage and a concise summary.

## B.2. Early Stopping Judgment Criteria

**Instruction Following.** Evaluate how well the outline follows the user's instructions for an outline. This includes adherence to the specified topic and scope, intended audience, purpose, constraints, required sections, level of detail, tone, and any formatting or length requirements. The evaluation should also check outline-specific expectations such as a clear hierarchical structure (e.g., H1/H2/H3 or bullet levels), logical ordering, consistent granularity across sections, correct numbering if requested, and inclusion of all required components (e.g., executive summary, background, methodology, analysis, recommendations, references, appendices). Penalize missing required elements, inclusion of prohibited items, incorrect scope or level of detail, or deviation from the requested format.
**Depth.** Assess the comprehensiveness and analytical depth of the outline. High-depth outlines move beyond broad headings to include specific subpoints, key arguments, mechanisms or causal drivers, assumptions and uncertainties, methods to be used, metrics, and success criteria. They indicate sequencing and logic (what builds on what), note dependencies and open questions, and identify where evidence, examples, and visuals will be integrated. Shallow outlines list generic topics without meaningful substructure, rationale, or analytical scaffolding.
**Balance.** Evaluate the fairness and objectivity of the outline. Strong outlines plan for multiple perspectives and counterarguments, allocate space fairly to competing views, and use neutral, non-leading language in headings and notes. Where issues are controversial or multi-faceted, the outline should explicitly include sections for trade-offs, limitations, and counter-evidence. Poor outlines display bias, give disproportionate space to one side without justification, or omit salient opposing views.
**Breadth.** Evaluate how many distinct and relevant subtopics, perspectives, or contexts the outline covers, while staying focused on the brief. Excellent outlines include appropriate dimensions such as historical context, legal or regulatory considerations, economic or market factors, technical or operational aspects, ethical implications, social or cultural impacts, geographic or comparative analysis, stakeholder perspectives, risks and limitations, and implementation pathways. Coverage should be wide-ranging yet purposeful; simply presenting two sides of a debate is insufficient, and irrelevant tangents should be avoided.
**Support.** Evaluate the outline's evidentiary scaffolding and sourcing plan. Providing source URLs somewhere in the outline (for example, in a references section or via inline citations) is the minimum requirement; if no section provides source URLs, the score must be zero. Factual accuracy is necessary but not sufficient. Higher-quality outlines explicitly attribute planned factual claims to verifiable sources (such as peer-reviewed articles, government databases, or reputable news organizations) with traceable citations including author or outlet, date, and URL. Quantitative points specify concrete datasets or reports, time frames, and comparative benchmarks. Qualitative points identify concrete examples or case studies, clearly linked to the argument, with sources. Sources should be credible and balanced; cherry-picking or omission of clearly relevant counter-evidence is penalized. Original synthesis should build on cited material, not replace it.
**Insightfulness.** Assess how insightful and practically useful the outline is. Excellent outlines go beyond common templates by offering original structure or framing, highlighting non-obvious but relevant connections, and sequencing sections to surface key insights efficiently. Recommendations and proposed analyses are concrete and actionable, clearly indicating what will be done, where it will appear, and how outcomes will be measured. Strong outlines call out specific real-world examples or comparator cases (who did what, when, what outcomes were observed, and how they were measured) and propose suitable exhibits such as tables, charts, or frameworks with a clear analytical purpose. Vague, generic, or purely aspirational notes cannot score highly.

## B.3. Algorithms for DualGraph

**Function Definitions.** Algorithm 1 use the following subroutines. We describe each function's purpose, inputs/outputs, and key implementation details to ensure reproducibility.

- CREATEOUTLINE($q_0$) → $OG_0$. Initializes an outline graph (tree-structured report plan) from the root query $q_0$ using

---

**Algorithm 1** Iterative Workflow of DualGraph

---

**Input:** $q_0, T$       **Output:** $\mathcal{R}$
**Defs:** OG: Outline Graph;  KG: Knowledge Graph;  $\mathcal{Q}$: query set;  $\mathcal{D}$: retrieved evidence
$\text{OG}_0 \leftarrow \text{CREATEOUTLINE}(q_0)$
$\mathcal{Q}_0 \leftarrow \text{GENFROMOG}(\text{OG}_0)$
$\mathcal{D}_0 \leftarrow \text{SEARCH}(\mathcal{Q}_0);\ \ \text{KG}_0 \leftarrow \text{BUILDKG}(\mathcal{D}_0)$
**for** $t = 0$ **to** $T - 1$ **do**
   $\mathcal{Q}_t^{\text{KG}} \leftarrow \text{GENFROMKG}(\text{KG}_t, \text{OG}_t)$
   **if** $t > 0$ **then**
      $\mathcal{Q}_t^{\text{OG}} \leftarrow \text{GENFROMOG}(\text{OG}_t)$
   **end if**
   $\mathcal{Q}_{t+1} \leftarrow \text{DEDUP}(\mathcal{Q}_t^{\text{KG}} \cup \mathcal{Q}_t^{\text{OG}})$
   $\mathcal{D}_{t+1} \leftarrow \text{SEARCH}(\mathcal{Q}_{t+1})$
   $\text{KG}_{t+1} \leftarrow \text{UPDATEKG}(\text{KG}_t, \mathcal{D}_{t+1})$
   $\text{OG}_{t+1} \leftarrow \text{UPDATEOG}(\text{OG}_t, \text{KG}_{t+1}, \mathcal{D}_{t+1})$
   **if** $\text{EARLYSTOP}(\text{OG}_{t+1})$ **then**
     **break**
   **end if**
**end for**
$\mathcal{D} \leftarrow \bigcup_{t=0}^{T} \mathcal{D}_t$
$\mathcal{R} \leftarrow \text{WRITEREPORT}(\text{OG}_{\text{final}}, \mathcal{D})$

---

an LLM (Prompt: `CREATE OUTLINE` in Appendix I.1). The output is a numbered hierarchical outline with no citations at initialization.

- GENFROMOG$(\text{OG}_t) \rightarrow \mathcal{Q}_t^{\text{OG}}$. Generates search queries from the current outline by targeting under-supported parts of $\text{OG}_t$ (i.e., nodes/lines without `<citation>id_*</citation>` markers) and avoiding redundancy with previously executed (and pending) queries (Prompt: `GENERATE SEARCH QUERIES` in Appendix I.2). The output is a set of web-search-friendly queries.

- SEARCH$(\mathcal{Q}_t) \rightarrow \mathcal{D}_t$. Executes web retrieval and evidence acquisition for a query set $\mathcal{Q}_t$ following the pipeline in Appendix B.1: (i) Bing top-$N$ URL retrieval per query; (ii) URL-level LLM filtering based on title/snippet; (iii) crawling and parsing; (iv) evidence-level LLM extraction conditioned on the originating query, retaining only useful pages. The system maintains a global URL de-duplication set across all iterations. The output $\mathcal{D}_t$ is an evidence bank of citable evidence units with unique IDs.

- BUILDKG$(\mathcal{D}_0) \rightarrow \text{KG}_0$. Constructs an initial knowledge graph from the initial evidence bank. We use an LLM to extract (i) core-entity nodes, (ii) concept nodes, and (iii) directed relational edges grounded by evidence IDs (Prompt: `EXTRACT KNOWLEDGE NODES` in Appendix I.5). Each extracted edge stores its supporting evidence IDs.

- UPDATEKG$(\text{KG}_t, \mathcal{D}_{t+1}) \rightarrow \text{KG}_{t+1}$. Updates the knowledge graph with newly acquired evidence. Concretely, it performs: (1) LLM-based node/edge extraction and grounding with evidence IDs (Prompt: `EXTRACT KNOWLEDGE NODES`); (2) semantic canonicalization by merging near-duplicate concept nodes and redirecting incident edges (Prompt: `MERGE KNOWLEDGE NODES` in Appendix I.6); (3) semantic clustering of nodes using embedding similarity; (4) community detection on the evolving KG using the Leiden algorithm. Evidence attachment is edge-centric: when a relation is re-observed, we union its evidence-ID set.

- GENFROMKG$(\text{KG}_t, \text{OG}_t) \rightarrow \mathcal{Q}_t^{\text{KG}}$. Performs KG-driven gap discovery and converts it into concrete search queries. First, it constructs a set of candidate *search chains* from $\text{KG}_t$ using the enrichment and exploration procedures in Appendix C (weakly supported existing edges for ENRICH; hypothesized missing relations for EXPLORE). Then, given the root query, current outline, current KG, and candidate chains, an LLM selects up to a fixed budget of chains and generates aligned search queries (Prompt: `KG CHAIN SELECTION & QUERY GENERATION` in Appendix I.3). The output is a set of search queries specialized to fill KG gaps while respecting the report context from $\text{OG}_t$.

- DEDUP$(\mathcal{Q}_t^{\text{KG}} \cup \mathcal{Q}_t^{\text{OG}}) \rightarrow \mathcal{Q}_{t+1}$. Deduplicates and normalizes candidate queries by removing exact duplicates and near-duplicates (e.g., trivial paraphrases) to avoid redundant retrieval within and across iterations.

- UPDATEOG($\text{OG}_t, \text{KG}_{t+1}, \mathcal{D}_{t+1}$) $\to \text{OG}_{t+1}$ for DualGraph, and UPDATEOG($\text{OG}_t, \mathcal{D}_t$) $\to \text{OG}_{t+1}$ for DualGraph w/o KG. Revises the outline structure and maintains citation grounding after each retrieval round. The update may add/remove/re-scope sections, and attaches evidence IDs to supported nodes/lines. We enforce *citation persistence*: when a section is edited, split, merged, or moved, its existing citation IDs are preserved and propagated to the most semantically corresponding descendant/parent nodes to prevent citation loss across iterations. In DualGraph, KG context can also be provided to the LLM to encourage globally consistent outline edits.

- EARLYSTOP($\text{OG}_{t+1}$) $\to \{\texttt{true}, \texttt{false}\}$. Determines whether iterative exploration should terminate by scoring the current outline along instruction following, depth, breadth, balance, support, and insightfulness (criteria in Appendix B.2). Termination triggers when all dimension scores exceed preset thresholds, preventing redundant searching.

- WRITEREPORT($\text{OG}_{\text{final}}, \mathcal{D}$) $\to \mathcal{R}$. Generates the final report section-by-section following $\text{OG}_{\text{final}}$. For each outline node/section $s_i$, we retrieve only the evidence units whose IDs appear in its citation set $C_i$ and synthesize the section with an LLM (Prompt: WRITE SECTION BY OUTLINE in Appendix I.4), citing only the provided evidence IDs.

## C. Search Chain Construction and Filtering

DualGraph constructs *SearchChains* from the Knowledge Graph (KG) to drive subsequent search queries. The goal of this process is to provide *prioritized exploration cues* for retrieval: on the one hand, to enrich existing relations with insufficient evidence coverage (**Enrich**); on the other hand, to explore high-value or high-confidence missing relations (**Explore**).

**Quota Allocation**  Given a total budget of $N$ SearchChains, we allocate $\lfloor N/4 \rfloor$ chains to the Enrich category. The remaining three Explore types are each assigned $M = \lfloor N/4 \rfloor$ chains. Any unused slots caused by insufficient candidates in one category are filled by remaining candidates from other categories according to a predefined priority order.

### C.1. Enrichment Search Chain (Existing Edges)

The *enrichment search chain* targets relations that already exist in the KG but are weakly supported by evidence. Given the current KG $(V, E)$, we first construct an enrichment candidate pool by filtering edges with low evidence coverage:

$$\text{pool}_{\text{enrich}} = \{\, e \in E \mid \text{evidence\_count}(e) \leq \text{enrich\_evidence\_threshold} \,\}. \tag{1}$$

We then rank candidate edges by a composite score that favors (i) edges involving core entities and (ii) edges with higher cross-community exposure, while down-weighting edges that already have more evidence.

**(1) Cross-community exposure.** We define a node-level bridging score

$$\text{bridging\_score}(v) = \frac{\text{cross\_neighbors}(v)}{\max(\deg(v), 1)}, \tag{2}$$

and compute the cross-community exposure of an edge $(u, v)$ as

$$\text{cross\_community}(u, v) = \tfrac{1}{2}\big(\text{bridging\_score}(u) + \text{bridging\_score}(v)\big). \tag{3}$$

**(2) Core-entity importance.** To prioritize relations that are central to the user query, we use

$$\text{node\_importance}(u, v) = \begin{cases} 1.0 & \text{if both } u \text{ and } v \text{ are core entities,} \\ 0.5 & \text{if exactly one of } u, v \text{ is a core entity,} \\ 0.0 & \text{if neither } u \text{ nor } v \text{ is a core entity.} \end{cases} \tag{4}$$

**(3) Composite enrichment score.** The final enrichment score is

$$\text{Score}_{\text{enrich}}(u, v) = \frac{1 + \text{node\_importance}(u, v) + \text{cross\_community}(u, v)}{1 + \text{evidence\_count}(u, v)}. \tag{5}$$

**Ranking.** We adopt a two-stage strategy to ensure edges with no evidence are handled first:

1. rank all candidates with $\text{evidence\_count}(u, v) = 0$ before any others;

2. rank remaining candidates by $\text{Score}_{\text{enrich}}(u, v)$ in descending order.

### C.2. Exploratory Search Chain (Missing Edges)

The *exploratory search chain* targets potentially important relations that are *not yet present* in the current KG. To balance high-confidence completion and high-value exploration, we generate missing-edge candidates using three complementary signals.

**Type I: CoreEntity–Concept candidates.** We enumerate all unconnected core entity–concept pairs and treat them as candidates if $(c, k) \notin E$:

$$\text{pool}^{(1)}_{\text{explore}} = \{(c, k) \mid c \in \mathcal{C}, \ k \in \mathcal{K}, \ (c, k) \notin E\}. \tag{6}$$

Candidates are ranked by semantic similarity $\text{sim}(c, k)$ in descending order, and the Top-$M$ pairs are selected as exploratory chains.

**Type II: SBM-based cross-community candidates.** We use a Stochastic Block Model (SBM) to propose missing edges across communities. For communities $(c_i, c_j)$, the inter-block probability is estimated with Laplace smoothing:

$$B[c_i][c_j] = \frac{\text{actual\_edges}(c_i, c_j) + \alpha}{\text{possible\_pairs}(c_i, c_j) + 2\alpha}, \quad \alpha = 0.1, \tag{7}$$

where

$$\text{possible\_pairs}(c_i, c_j) = \begin{cases} n_i n_j, & c_i \neq c_j, \\ \frac{n_i(n_i - 1)}{2}, & c_i = c_j. \end{cases} \tag{8}$$

We enumerate all node pairs $(i, j)$ with $\text{comm}(i) \neq \text{comm}(j)$ and $(i, j) \notin E$ as the Type-II pool. Within the Type-II quota, we select candidates by combining:

- **Probability-ranked half:** $p_{ij} = B[\text{comm}(i)][\text{comm}(j)]$,

$$p_{ij} = B[\text{comm}(i)][\text{comm}(j)], \tag{9}$$

  selecting the Top-$\lfloor M/2 \rfloor$ edges by descending $p_{ij}$.

- **Uncertainty-ranked half:** entropy of the Bernoulli parameter,

$$H(p_{ij}) = -p_{ij} \log_2 p_{ij} - (1 - p_{ij}) \log_2(1 - p_{ij}), \tag{10}$$

  selecting the Top-$\lfloor M/2 \rfloor$ edges by descending $H(p_{ij})$.

(When the quota is odd, flooring is applied and remaining slots are filled by priority.)

**Type III: Structural-hole candidates.** To uncover missing relations that may connect distinct knowledge communities, we exploit structural-hole signals. We define intra-community degree as

$$\text{deg}_{\text{in}}(v) = |\{u \in \text{neighbors}(v) \mid \text{comm}(u) = \text{comm}(v)\}|. \tag{11}$$

Within each community, we select representative nodes:

- **Bridge nodes:** among nodes with $\text{cross\_neighbors}(v) > 0$, select the Top-3 by $\text{bridging\_score}(v)$;

- **Hub nodes:** select the Top-2 by $\text{deg}_{\text{in}}(v)$, excluding Bridge nodes;

- **Representative node (Rep):** select the node with the highest $\text{deg}_{\text{in}}(v)$.

For any distinct community pair $(c_1, c_2)$, we pair Bridge/Hub nodes from $c_1$ with the Rep node from $c_2$; if the corresponding edge does not exist, it is added as a Type-III exploratory candidate.

## D. Formal Analysis of Search Space, Coverage, and Convergence

Deep research systems must decide what to search next at each iteration. Let $\mathcal{Q}_t$ denote the candidate query space at iteration $t$. Without structural constraints, $\mathcal{Q}_t$ is freely generated by the LLM and can be arbitrarily large.

**Search space under KG constraints.** DualGraph decomposes $\mathcal{Q}_t$ into two bounded subsets:

- **Enrich space:** $\mathcal{Q}_t^{\mathrm{enrich}} \subseteq \{e \in E_t \mid \text{evidence-count}(e) \leq \tau_{\mathrm{enrich}}\}$, i.e., existing but under-evidenced edges in the KG. Its upper bound is $|E_t|$ (current edge count), and after ranking by the composite enrichment score $\mathrm{Score}_{\mathrm{enrich}}(u, v) = \frac{1+\text{node-importance}(u,v)+\text{cross-community}(u,v)}{1+\text{evidence-count}(u,v)}$, only the top-$\lfloor N_{\mathrm{KG}}/4 \rfloor$ candidates are selected. This subset addresses "known but under-evidenced" gaps—strengthening existing relations.

- **Explore space:** $\mathcal{Q}_t^{\mathrm{explore}} \subseteq (V_t \times V_t) \setminus E_t$, i.e., node pairs not yet connected in the KG. Although theoretically $O(|V_t|^2)$, triple filtering retains only top-$\lfloor N_{\mathrm{KG}}/4 \rfloor$ candidates per type: Type I (semantic similarity) addresses "should exist but undiscovered" links; Type II (SBM cross-community probability/entropy) addresses "implicit cross-topic links"; Type III (structural hole signals) addresses "gaps between topic islands".

Thus, the total number of KG-driven search chains per iteration is strictly bounded by $N_{\mathrm{KG}}$, and the total search chain count is $N = N_{\mathrm{OG}} + N_{\mathrm{KG}}$.

**Monotonic coverage growth and search space shrinkage.** Critically, each iteration covers new ground without redundantly re-exploring previously resolved gaps:

- For Enrich, once an edge accumulates sufficient evidence (evidence-count$(e) > \tau_{\mathrm{enrich}}$), it exits the candidate set permanently.

- For Explore, once a node pair is investigated, the KG update either adds it to $E_{t+1}$ or reprioritizes it under the new topology, preventing redundant queries.

Since newly retrieved evidence simultaneously updates the KG and OG, the set of resolved knowledge gaps is monotonically non-decreasing across iterations. The effective residual search space therefore tends to shrink over time, transforming exploration from "guessing what's missing" to "graph topology indicating what's missing."

**Convergence.** The bounded per-iteration search budget and non-repetitive gap resolution keep the selected search chains finite and reduce redundant exploration, encouraging convergence in practice. Greedy selection of highest-scoring gaps further concentrates gains in early iterations.

## E. Cost Analysis

To quantify the cost of KG maintenance, we compare DualGraph with open-source baselines on a 20-case subset of DeepResearch Bench. Table 5 reports the token usage and average runtime for each method.

*Table 5.* Token usage and runtime comparison on a 20-case subset of DeepResearch Bench.

| Method | Input (K) | Output (K) | Total (K) | Avg. Time (min) |
|---|---|---|---|---|
| DualGraph | 2,301 | 189 | 2,490 | 36.75 |
| DualGraph w/o KG | 1,661 | 137 | 1,798 | 30.68 |
| WebWeaver | 1,404 | 132 | 1,536 | 28.80 |
| Langchain-DR | 833 | 64 | 897 | 13.40 |
| Tongyi-DR | 335 | 15 | 350 | 7.10 |
| Qwen-Agent | 425 | 21 | 446 | 9.70 |

From Table 5, we acknowledge that DualGraph incurs additional computational cost compared to other open-source methods. However, this cost enables our system's multiple modules to generate better search queries, perform more comprehensive exploration, and collect more complete information sources. This design emphasizes understanding query directions, which

*Table 6.* Per-module token and runtime breakdown of DualGraph and DualGraph w/o KG.

| Module | DualGraph Tokens (K) | DualGraph Time (min) | w/o KG Tokens (K) | w/o KG Time (min) |
|---|---|---|---|---|
| Generate Queries | 48 | 0.43 | 26 | 0.27 |
| Judge Terminal | 76 | 1.01 | 79 | 1.39 |
| Maintain Outline | 127 | 2.08 | 96 | 1.97 |
| Maintain KG | 224 | 3.48 | — | — |
| WebSearch/Crawl | 1,674 | 26.08 | 1,308 | 23.46 |
| Write Paper | 341 | 3.67 | 289 | 3.59 |
| Total | 2,490 | 36.75 | 1,798 | 30.68 |

is the core driver of our system's superior performance. Moreover, as shown in Table 1, DualGraph achieves the highest RACE among these open-source methods, indicating that the additional cost is associated with meaningful performance gains.

From Table 6, KG maintenance accounts for only $9\%$ of total tokens, while more comprehensive web search and crawling dominates at $67\%$, which is an unavoidable cost for high-quality deep research. This shows that DualGraph's overhead is primarily driven by more comprehensive exploration, not by the KG component itself. In the report generation domain, the primary goal is to deliver high-quality, trustworthy, and verifiable reports; our current design prioritizes this high standard.

## F. KG Extraction Reliability

We acknowledge that LLM-based extraction of graph entities and relations inevitably introduces errors. We analyzed all 6,014 entity-relation triplets extracted at Iter 1 across 100 cases in DeepResearch Bench: only $2.4\%$ are anomalous, consistent with recent studies showing high accuracy of LLMs on structured information extraction tasks. Table 7 provides the detailed breakdown.

*Table 7.* Anomaly analysis of KG triplet extraction at Iter 1 across 100 cases (6,014 triplets total).

| Category | % of All Edges | Definition |
|---|---|---|
| Loosely Relevant | 1.48% | The extracted relation is related but not directly stated in the evidence |
| Hallucination | 0.65% | The triplet has no grounding in the evidence |
| Semantic Bias | 0.27% | The relation direction or meaning is subtly distorted |
| Total | 2.40% | |

Only $0.12\%$ (7 triplets) of all triplets are both anomalously extracted *and* used to drive search queries before correction. The remaining anomalous extractions are effectively mitigated by DualGraph's built-in self-correction mechanism:

1. Although *Loosely Relevant* extractions are not directly supported in the current iteration, they serve two purposes: helping generate new search queries and, in many cases, being later validated by newly retrieved evidence confirming the triplet is factually correct.

2. DualGraph performs merge or update operations on the KG at each iteration (see Methodology, step ❸), which self-corrects errors including *Hallucination* and *Semantic Bias*. If newly retrieved evidence contradicts an existing relation, the system automatically corrects or deletes the erroneous edge. For example, in Case 79, the initial extraction was "(Misrepresentation of LGBTQ+ in MENA media)-[exacerbates]->(Legal barriers for transgender people)", which reverses the true causality (*Semantic Bias*). After retrieving new evidence, the system automatically corrected the relation to "-[results from]->".

## G. Human Evaluation of LLM-as-a-Judge Reliability

To validate the reliability of LLM-based evaluation, we conducted a fine-grained human evaluation study. We randomly sampled 5 evaluation cases each from OpenAI Deep Research and DualGraph on DeepResearch Bench, yielding 10

cases covering 254 rubric criteria. Each case includes the evaluated report, the reference benchmark report, and the LLM judge's per-criterion analysis and scores across four dimensions. Five human annotators with research experience independently reviewed each rubric criterion. After reading both reports (anonymized), annotators were instructed to mark each LLM-assigned score and its corresponding reasoning as "reasonable" or "unreasonable".

The annotation results are summarized as follows:

1. The **all-annotators-agree rate**, defined as the proportion of criteria where all five annotators unanimously endorsed the LLM's score and reasoning, is 88.6% (225/254).

2. The **per-judgment agreement rate**, defined as the fraction of individual human judgments (each annotator $\times$ each criterion) rated as reasonable, is 97.5% (1,238/1,270).

3. **Inter-annotator agreement** is high, with Gwet's AC1 at 0.95.

Together, these results provide strong evidence that the LLM-as-a-Judge protocol used in our evaluation produces reliable and consistent assessments.

## H. Case Trajectory



### CASE TRAJECTORY IN DUALGRAPH.

#### Root Question

**What are the investment philosophies of Duan Yongping, Warren Buffett, and Charlie Munger?**

#### Step 1 (Iter 0) - Init OG

[OG]

```
Comparative Analysis of Investment Philosophies: Duan Yongping, Warren Buffett, and Charlie Munger
1. Introduction
    1.1 Purpose and Scope
    1.2 Overview of Subject Investors
    1.3 Structure of the Report
2. Duan Yongping's Investment Philosophy
    2.1 Background and Influences
    2.2 Core Principles
    2.2.1 Value Investing Approach
        a. Focus on intrinsic value and margin of safety
        b. Importance of long-term holding
    2.2.2 Emphasis on Simplicity and Patience
        a. Avoiding complex investments
        b. Waiting for clear opportunities
    2.2.3 Risk Management Strategies
        a. Selective investment decisions
        b. Use of cash reserves
    2.3 Notable Investments and Strategies
    2.4 Impact on Chinese Investment Community
[OMITTED]
```

#### Step 2 (Iter 0) - GenFromOG

[SQ]
SQ1: influence of Duan Yongping's investment philosophy on Chinese retail investors
SQ2: Duan Yongping's approach to risk management in volatile markets
SQ3: examples of Duan Yongping's notable investments and their outcomes
SQ4: how Warren Buffett calculates intrinsic value in practice
[OMITTED]

#### Step 3 (Iter 0) - Search

[Evidence]
Evidence1: title: The Investment Philosophy of Duan Yongping • Honwhy's website url: https://honwhy.wang/duanyongping/ content: **Summary**: The webpage outlines Duan Yongping's investment philosophy and notes that his principles have inspired a generation of entrepreneurs and investors in China. This directly addresses the influence of his philosophy on Chinese retail investors, emphasizing the spread of his values of simplicity, patience, and rational decision-making. **Original content**: A journey into the mind of China's legendary value investor - Duan Yongping, whose principles of simplicity, patience, and clarity inspired a generation of entrepreneurs and investors. Duan Yongping's philosophy emphasizes rationality, patience, and focus - a Chinese reflection of Buffett's wisdom with his own Eastern restraint and practicality.
**[other evidence items retrieved; omitted for clarity]**

#### Step 4 (Iter 0) - BuildKG



**[KG]**

### Step 5 (Iter 1) - GenFromKG

**[SQ]**
SQ1: How has Duan Yongping adapted value investing principles for emerging markets like China?
SQ2: How does Warren Buffett tailor his investment philosophy when applying value investing principles to emerging markets?
SQ3: How does Charlie Munger approach the adaptation of investment principles to emerging markets?
**[OMITTED]**

### Step 6 (Iter 1) - Search

**[Evidence]**
Evidence1: title: Interview with Duan Yongping: 50 Key Points on Investment, Culture, and ... url: https://www.gate.com/crypto-wiki/article/interview-with-duan-yongping-50-key-points-on-investment-culture-and-education-20260108 content: **Summary**: The webpage provides direct insights into Duan Yongping's risk management approach by emphasizing deep understanding of business models, cash flow analysis, and long-term patience over speculation. He manages investment risk by focusing on market maturity, preferring established companies with proven models and avoiding hype-driven emerging sectors. His philosophy centers on rational, informed decision-making and vigilance toward market cycles-principles that align with leveraging latecomer advantage and market maturity to reduce risk. **Original content**: Duan Yongping emphasized that investment is conceptually simple but challenging to execute in practice. According to his philosophy, successful investing requires a deep understanding of a company's business model and the ability to project future cash flows accurately. This approach aligns with value investing principles popularized by Warren Buffett, with whom Duan has maintained a long-standing relationship.
**[other evidence items retrieved; omitted for clarity]**

### Step 7 (Iter 1) - UpdatedKG

### Step 8 (Iter 1) - UpdatedOG

**[OG]**

```
Comparative Analysis of Investment Philosophies: Duan Yongping, Warren Buffett, and Charlie Munger
1.Introduction
    1.1 Purpose and Scope
    1.2 Overview of Subject Investors
    1.3 Structure of the Report
2.Duan Yongping's Investment Philosophy
    2.1 Background and Influences
    2.2 Core Principles
        2.2.1 Value Investing and Intrinsic Value
            a. Investing is buying a company, not just a stock; focus on business fundamentals and future
            ↪  cash flows <citation>id_29, id_30, id_31, id_32, id_33, id_34</citation>
            b. Margin of safety is rooted in deep understanding rather than simply buying cheap
            ↪  <citation>id_32, id_34</citation>
            c. Focusing on undervalued opportunities and qualitative discounted cash flow estimation
            ↪  <citation>id_34</citation>
        2.2.2 Simplicity and Clarity
            a. Maintain a "Do Not Do" list and operate strictly within one's circle of competence
            ↪  <citation>id_28, id_29, id_30, id_34</citation>
            b. Avoid complex, speculative, or poorly understood investments <citation>id_28, id_29, id_30,
            ↪  id_31, id_34</citation>
            c. Require business model clarity and transparency; invest only in understandable enterprises
            ↪  <citation>id_29, id_30, id_33, id_34</citation>
        2.2.3 Emotional Discipline and Patience
            a. Importance of rational, disciplined analysis over emotional reactions to market volatility
            ↪  <citation>id_30, id_33</citation>
            b. Prioritizing long-term holding and compounding returns over short-term market movements
            ↪  <citation>id_29, id_30, id_31, id_32, id_33, id_34</citation>
        2.2.4 Risk Management and Trust
            a. Adapting value investing to emerging markets: due diligence, trustworthiness, strict "Don't
            ↪  Do List" for risk mitigation <citation>id_26, id_28, id_34</citation>
            b. Only invest in trustworthy ventures and avoid those with misaligned values <citation>id_26,
            ↪  id_33</citation>
            c. Preserving capital by avoiding speculation and high-risk/immature markets <citation>id_26,
            ↪  id_31, id_34</citation>
        2.2.5 Value Alignment and Foundational Philosophy
            a. Invest in companies with clear, steadfast core philosophies and value alignment
            ↪  <citation>id_29, id_33</citation>
            b. Company culture and values fundamentally drive business sustainability <citation>id_29,
            ↪  id_34</citation>
        2.2.6 Opportunity Cost as a Decision Filter
            a. Evaluate choices against the best available alternative; reject "okay" opportunities when a
            ↪  better one exists <citation>id_35</citation>
            b. Treat opportunity cost as an implicit "Do Not Do" rule in capital allocation
            ↪  <citation>id_36</citation>
```

**[OMITTED]**

### Step 9 (Iter 2) - GenFromKG

SQ1: Impact of Duan Yongping's "Do Not Do" list on portfolio risk and return in emerging markets How Duan Yongping incorporates qualitative discounted cash flow estimation in investment decisions in Chinese companies

SQ2: Historical evolution of the teacher-student relationship between Warren Buffett and Benjamin Graham and its effect on Buffett's investment style
**[OMITTED]**

---

### Step 10 (Iter 2) - GenFromOG

SQ1: How does Charlie Munger adapt investment principles for emerging markets, and what evidence supports this? What strategies does Warren Buffett use to apply his investment principles in emerging markets, and how does this differ from his U.S.-centric approach?
SQ2: Is there evidence of Duan Yongping utilizing a multidisciplinary approach to investing, and how does it compare to Charlie Munger's method?
SQ3: What were the key factors and influences behind Warren Buffett's shift from 'cigar-butt' investing to quality investing?
**[OMITTED]**

---

### Step 11 (Iter 2) - Search

**[Evidence]**
Evidence1: title: The Way is Simple: Analysis of Duan Yongping's Investment Wisdom url: https://en.shuziqushi.com/new224704.html content: **Summary**: The page is useful because it explains that Duan Yongping defines simplicity in investing as focusing on the fundamentals, avoiding speculation and leverage, investing only in businesses he truly understands, and maintaining patience through long-term, steady accumulation. He uses metaphors such as 'farming' versus 'hunting' to highlight the importance of a calm, rational, and simple approach, avoiding shortcuts and emotional decisions. His philosophy is about sticking to what one should be doing, playing the long game, and not being distracted by market noise or quick wins. **Original content**: In early 2025, Duan Yongping, a well-known investor and entrepreneur, went back to his alma mater, Zhejiang University, and shared some thoughts: Honestly, I want to make quick money, who doesn't? But its crucial to realize that it just doesn't happen. You still need to stick to what you should be doing. This came from his book Bao: Duan Yongpings Investment Questions and Answers, published in May 2025.
**[other evidence items retrieved; omitted for clarity]**

---

### Step 11 (Iter 2) - UpdatedKG

---

### Step 12 (Iter 2) - UpdatedOG

**[OG]**

```
Comparative Analysis of Investment Philosophies: Duan Yongping, Warren Buffett, and Charlie Munger

1. Introduction
    1.1 Purpose and Scope
    1.2 Overview of Subject Investors
    1.3 Structure of the Report
2. Duan Yongping's Investment Philosophy
    2.1 Background and Influences
        2.1.1 Intellectual Lineage and Relationship to Buffett and Munger <citation>id_60</citation>
            a. Duan as Buffett's student and firm practitioner of value investing
            b. Philosophical continuity and adaptation to Chinese context
    2.2 Core Principles
        2.2.1 Value Investing and Intrinsic Value
            a. Investing is buying a company, not just a stock; focus on business fundamentals and
            ↪ future cash flows <citation>id_29, id_30, id_31, id_32, id_33, id_34, id_50, id_51, id_56,
            ↪ id_59</citation>
            b. Margin of safety is rooted in deep understanding rather than simply buying cheap
            ↪ <citation>id_32, id_34, id_52, id_53, id_54</citation>
            c. Focusing on undervalued opportunities and qualitative discounted cash flow estimation
            ↪ <citation>id_34, id_54, id_56</citation>
            d. Preference for conservative, gross estimates and qualitative judgment over mathematical
            ↪ precision <citation>id_56</citation>
        2.2.2 Simplicity and Clarity
            a. Maintain a "Do Not Do" list and operate strictly within one's circle of competence
            ↪ <citation>id_28, id_29, id_30, id_34, id_50</citation>
            b. Avoid complex, speculative, or poorly understood investments; caution toward emerging
            ↪ technologies <citation>id_28, id_29, id_30, id_31, id_34, id_50, id_72</citation>
            c. Require business model clarity and transparency; invest only in understandable
            ↪ enterprises <citation>id_29, id_30, id_33, id_34, id_50</citation>
        2.2.3 Emotional Discipline and Patience
            a. Importance of rational, disciplined analysis over emotional reactions to market
            ↪ volatility; avoidance of FOMO <citation>id_30, id_33, id_50, id_58</citation>
            b. Prioritizing long-term holding and compounding returns over short-term market movements
            ↪ <citation>id_29, id_30, id_31, id_32, id_33, id_34, id_50, id_51, id_58</citation>
        2.2.4 Risk Management and Trust
            a. Adapting value investing to emerging markets: due diligence, trustworthiness, strict
            ↪ "Don't Do List" for risk mitigation <citation>id_26, id_28, id_34, id_50, id_51</citation>
            b. Only invest in trustworthy ventures and avoid those with misaligned values
            ↪ <citation>id_26, id_33, id_50, id_51</citation>
            c. Preserving capital by avoiding speculation, leverage, and high-risk/immature markets;
            ↪ maintain cash reserves for unknowns <citation>id_26, id_31, id_34, id_50, id_51, id_52,
            ↪ id_53, id_75</citation>
        2.2.5 Value Alignment and Foundational Philosophy
            a. Invest in companies with clear, steadfast core philosophies and value alignment
            ↪ <citation>id_29, id_33, id_50</citation>
            b. Company culture and values fundamentally drive business sustainability <citation>id_29,
            ↪ id_34, id_50, id_57, id_59</citation>
            c. Consumer orientation as the essence of good corporate culture <citation>id_57</citation>
        2.2.6 Opportunity Cost as a Decision Filter
```

```
                      a. Evaluate choices against the best available alternative; reject "okay" opportunities when
                  ↪   a better one exists <citation>id_35, id_50</citation>
                      b. Treat opportunity cost as an implicit "Do Not Do" rule in capital allocation
                  ↪   <citation>id_36, id_50</citation>
[OMITTED]
```

Step 13 - WriteReport

[Final Report]

# I. Prompt Template

## I.1. Create Outline

### PROMPT TEMPLATE: CREATE OUTLINE

```
- You are an expert research assistant specializing in building and iteratively refining recursive outlines
↪   for deep-dive reports.
- Your task is to **create** a detailed outline that includes sections, subsections, and key points to be
↪   covered in each part of the research paper.
- Refer to Section `Input Format and Interpretation` below to understand the input format.
- Refer to Section `Guidance of Creating a New Outline` below for detailed instructions on how to create the
↪   outline.
- Refer to Section `Output Requirement and Format Rules` below for the required output format.
# Input Format and Interpretation
- The user will provides ONLY a root query (a single research question or topic).
- Example Input for Create Request:
```
What are the mechanisms, biomarkers, and therapeutic strategies for early-stage Parkinson's Disease?
```
# Guidance of Creating a New Outline
- Derive a logical structure from the root query.
- Choose a clear report title.
- Build sections that cover background, key dimensions, evidence, implications, etc.
- Do not include citation IDs unless the user explicitly provides supporting evidence.
- Prioritize logical flow, research utility, and traceability to evidence.
- Avoid overly deep nesting; use deeper levels only when necessary for precision.
- The outline should be readable as a standalone roadmap of the final report.
# Output Requirement and Format Rules
- Your output must always be a clean, human-readable plain-text outline that can directly serve as the table
↪   of contents for the final report.
- The **first line** must be the final report's title (concise and descriptive).
- Starting from the **second line**, list every section and subsection **one per line**, using strictly
↪   decimal hierarchical numbering for titles only:
  - Top-level sections: 1., 2., 3., ...
  - Second level: 1.1, 1.2, 2.1, ...
  - Third level: 1.1.1, 1.1.2, 1.2.1, ...
  - Do not use Level 4 titles (e.g., 1.1.1.1 or a. as a heading).
- Optionally, immediately after a Level 3 section line (x.x.x), you may include content summary items that
↪   elaborate what the section will cover.
  - These are not headings: they are supporting content points.
  - Format them as: a., b., c., d., ..., one per line, without indentation.
  - Including these items is optional: if a Level 3 section is self-explanatory or covers a single idea,
  ↪   omit the a./b. lines entirely.
- **Do NOT use any indentation**. Hierarchy is conveyed solely by the numbering and line order.
- **Do NOT include any extra text**-no explanations, no markdown, no blank lines.
- Citations (if any) appear inline as: <citation>id_X, id_Y</citation> at the end of the relevant line. For
↪   example: 1. Hoehn and Yahr Scale Classifications <citation>id_2, id_6, id_9...</citation>
```

## I.2. Generate Search Queries

### PROMPT TEMPLATE: GENERATE SEARCH QUERIES

```
- You are an expert research assistant specializing in identifying knowledge gaps in a report outline and
↪   generating precise, search engine friendly queries to fill them.
- Your task is to analyze the provided outline, detect sections that lack citation markers (i.e., no
↪   <citation>id_...</citation>), and based on their content and context-formulate $QUERY_NUM new,
↪   high-value search queries that would help retrieve relevant evidence to support those under-researched
↪   parts.
- Refer to Section `Input Format and Interpretation` below to understand the input format.
- Refer to Section `Guidance` below for detailed instructions on how to generate effective search queries.
- Refer to Section `The Output Format` below for the required output format.
# Input Format and Interpretation
```

```
- The input you receive in each call (via the user prompt) will contain the following components:
- Current Outline: A plain-text hierarchical outline of the report, where:
  - The first line is the report title
  - Subsequent lines represent sections/subsections using standard numbering (e.g., I, 1., a., etc.)
  - Some lines may include inline citation markers like <citation>id_5</citation>; lines without such
  ↪  markers are considered not yet supported by evidence and are primary candidates for query generation
- Historical Search Queries (executed): A list (or block) of all search queries that have already been
↪  executed during this research process. These must be avoided or meaningfully differentiated to prevent
↪  redundant searches.
- Pending Search Queries (planned, NOT executed yet) (optional): A list of search queries that have been
↪  generated and are scheduled to be executed soon, but have NOT been executed yet. You should not describe
↪  them as already searched, but you must still avoid duplicating them; instead, generate complementary
↪  queries that fill other gaps.
# Guidance
- Prioritize outline lines that:
  - Make factual, causal, statistical, or comparative claims
  - Reference specific concepts, policies, effects, or populations
  - Appear in deeper subsections (e.g., under "Mechanisms" or "Case Studies")-these often need targeted
  ↪  evidence
- Ignore purely structural or navigational lines (e.g., "Introduction", "Conclusion", "Literature Review")
↪  unless they contain substantive uncited claims.
- When multiple uncited lines relate to the same theme, synthesize them into a single, comprehensive query
↪  rather than duplicating.
- Favor queries that combine concept + domain + outcome/measure (e.g., "impact of urban green space on
↪  adolescent cortisol levels 2020-2025") over broad topic queries.
- Use recent years (e.g., "since 2020") or specific regions (e.g., "in Southeast Asia") only if the outline
↪  implies recency or locality.
- If historical queries already cover a topic broadly, drill down: e.g., shift from "green space mental
↪  health" to "green space access and ADHD symptom reduction in urban teens".
- Each query should:
  - Target a concrete, unresolved question or claim in an uncited outline line
  - Include relevant context (e.g., population, timeframe, mechanism, geography) when implied by the outline
  - Avoid vague terms like "overview", "introduction", or "information about"
- Do not include:
  - Numbering (e.g., "1. ...")
  - Quotation marks, JSON, markdown, or any formatting
  - Explanations, headers, or extra text
- Queries must be novel relative to the historical executed search queries, and also non-overlapping with
↪  pending (planned) queries. If pending queries already target a theme, generate adjacent or deeper
↪  queries that complement them (e.g., narrower mechanism, different population, specific metrics, recent
↪  years).
# The Output Format
- Output exactly $QUERY_NUM search queries, each on its own line.
- Use natural, concise, and specific English phrases optimized for web search engines (e.g., Google,
↪  Semantic Scholar).
```

## I.3. KG Chain Selection & Query Generation

> ### PROMPT TEMPLATE: KG CHAIN SELECTION & QUERY GENERATION
>
> ```
> - You are an expert research strategist specializing in knowledge graph refinement.
> - Your task is to analyze the candidate explore/enrich chains from an iterative DeepResearch process, select
> ↪  up to $CHAIN_NUM chains that should be pursued next, and generate aligned search queries to strengthen
> ↪  the knowledge graph with evidence.
> - Refer to Section `Input Format and Interpretation` below to understand the input format.
> - Refer to Section `Guidance` below for detailed instructions on how to select chains and write queries.
> - Refer to Section `The Output Format` below for the required output format.
>
> # Input Format and Interpretation
> - The input you receive (via the user prompt) contains:
>   - Root Query: the core research question/topic.
>   - Current Outline: a plain-text hierarchical outline; lines may include evidence markers like
>   ↪  <citation>id_5</citation>. Lines without markers are under-supported.
>   - Knowledge Graph: current entities/concepts/relations; missing or weak relations indicate gaps.
>   - Search Query Language: the language to use for queries (e.g., en, zh).
>   - Candidate Explore Chains: a list of candidate chains; each chain includes:
>     - chain id
>     - type: enrich or explore
>     - relation pattern (Entity → Relation → Concept)
> - Chain types:
>   1) type = enrich: the relation exists in the KG but needs stronger/clearer evidence.
>   2) type = explore: the relation is hypothesized and not yet present/verified in the KG.
>
> # Guidance
> - Select chains by prioritizing:
>   1) Relevance: resolving it directly advances the Root Query.
>   2) Gap severity: it fills a critical missing/weak relation in the KG.
> ```

```
   3) Leverage: it reduces ambiguity across multiple outline sections or enables multiple downstream
   ↪  inferences.
   4) Comparative insight (when applicable): it supports meaningful differentiation or alignment among key
   ↪  entities.
- Prefer chains that address core definitional/causal relationships, mechanisms, criteria, trade-offs, or
↪  decision rules; de-prioritize peripheral facts.
- For each selected chain, generate one search query that:
   - is search-engine friendly and specific
   - explicitly names key entities/concepts (no vague pronouns)
   - expresses the relationship to verify (support/oppose/define/cause/measure/compare)
   - matches the requested Search Query Language
- If the outline has high-priority evidence gaps not covered by any chain:
   - select fewer than $CHAIN_NUM chains
   - use remaining query slots to create additional outline-driven queries
   - keep total search queries <= $CHAIN_NUM

# The Output Format
- Output exactly one line: a valid JSON object with two keys:
   - "chains": an array of selected chain ids (length <= $CHAIN_NUM)
   - "search queries": an array of search queries (same order as "chains"; may append outline-driven queries
   ↪  after chain-based queries; total length <= $CHAIN_NUM)
- Do not include any other text, explanations, markdown, comments, or trailing commas.
```

## I.4. Write Section by Outline

### PROMPT TEMPLATE: WRITE SECTION BY OUTLINE

```
- You are an expert academic writer specializing in synthesizing provided research evidence into a
↪  publication-ready report section.
- Your task is to write ONE self-contained Markdown section that strictly follows the provided "Current
↪  Section Outline" (headings and numbering), integrates all required content points, and cites ONLY the
↪  provided evidence.
- Output ONLY the Markdown for this section (no preamble, no meta-commentary, no references).

# Input Format and Interpretation
- The user input will include:
   - Root Query: the core research question (context only).
   - Report Title: the final report title (context only; do NOT print it).
   - Previous Sections (optional): earlier Markdown text for continuity (do NOT repeat it).
   - Current Section Outline: the exact outline snippet you must write now.
      - Numbered lines (e.g., 2., 2.1, 2.1.1) are headings you MUST reproduce exactly.
      - Lettered lines (a., b., c., ...) are NOT headings; they are required content points for the
      ↪  immediately preceding numbered heading.
   - Supporting Evidence: evidence nodes with numeric IDs; these are the ONLY sources you may cite.

# Writing Rules (Strict)
- Headings:
   - Convert outline numbering to Markdown headings:
      - `1.`    -> `## 1. ...`
      - `1.1`   -> `### 1.1 ...`
      - `1.1.1` -> `#### 1.1.1 ...`
   - Preserve the exact wording and numbering of EVERY numbered heading from the outline.
   - Do NOT add, remove, merge, rename, or renumber headings.
   - Do NOT create any Level-4+ headings beyond `####` (no `#####`).
   - Lettered items (a./b./c./d.) must NEVER become headings (e.g., never write `#### 3.2.d`).
- Content integration:
   - Fully address every lettered requirement (a./b./c./...) in the correct location under the immediately
   ↪  preceding numbered heading.
   - You may use paragraphs, bullet lists, or tables to present structured comparisons, but keep the prose
   ↪  concise and academic.
   - Maintain continuity with "Previous Sections" by avoiding repeated explanations; only restate minimal
   ↪  context when needed for readability.
   - If the outline requests items you cannot support with the provided evidence, write them cautiously as
   ↪  analysis/synthesis without introducing new factual claims, and do not cite.
- Evidence and citation:
   - Cite when a sentence is supported by the provided evidence.
   - Citation format: use square brackets with numeric IDs only, e.g., [3] or [2,7].
   - Place citations at the end of the supported sentence (before the period).
   - Do NOT put citations in headings.
   - Do NOT cite anything not present in the Supporting Evidence; do NOT fabricate sources or IDs.
   - If multiple IDs support the same claim, list them in ascending order in one bracket, e.g., [2,7,9].
   - Keep claim-to-citation scope tight: do not bundle unrelated claims into one sentence with one citation.

# Output Requirements
- Output ONLY the Markdown content for the current section.
- Include all headings present in "Current Section Outline" and corresponding body text.
- Do NOT output the report title, any extra sections, or a references/bibliography.
```

## I.5. Extract Knowledge Nodes

---

**PROMPT TEMPLATE: EXTRACT KNOWLEDGE NODES**

```
– You are a knowledge synthesis expert. Your task is to construct a structured Knowledge Graph from the
↪  provided textual evidence.
– Extract knowledge relationships from the textual evidences (EN#) and map each evidence statement (EN#) to
↪  the relevant knowledge edges.
– Refer to Section `Input Format and Interpretation` below to understand the input format.
– Refer to Section `Guidelines for Knowledge Edge Extraction` below for detailed instructions on how to
↪  extract knowledge edges.
– Refer to Section `Output Requirement and Format Rules` below for the required output format.

# Input Format and Interpretation
– The input consists of:
  – Root Query: topic of this deep research report.
  – Search Query: the search query used to obtain the target evidence.
  – Numbered evidence statements: EN1: ..., EN2: ..., etc.
    – Each EN# is a self-contained piece of textual evidence (sentence or short paragraph).
  – Current Knowledge Graph (optional):
    – If not provided, this is the initial extraction step.
    – If provided, it includes existing nodes/edges from previous turns; you must reuse them and avoid
    ↪  duplication.

# Guidelines for Knowledge Edge Extraction
1) Knowledge Node Definition (two types)
– Core Entity Nodes:
  – Primary subjects of the Root Query; they emerge from context (not pre-defined).
  – They should be concrete entities (e.g., people, companies, organizations), not abstract topics.
  – NEVER create derivative nodes for a core entity (e.g., do not create "Ping An's Growth Strategy" if
  ↪  "Ping An" exists).
– Concept Nodes:
  – Other concepts/attributes/entities related to the core entities.
  – MUST connect (directly or indirectly) to at least one Core Entity node.

2) Knowledge Edge Definition
– Each edge is a directed relationship between two nodes.
– Each edge must be directly supported by at least one EN#.
– Use concise, natural-language relation names (e.g., "advocates", "rejects", "defines", "has attribute",
↪  "prioritizes").
– Avoid overly generic relations like "related to" unless unavoidable.

3) Edge-Centric Extraction Rule
– Decide the EDGES first; nodes exist only to support meaningful edges.
– Prioritize edges where at least one endpoint is a Core Entity node.
– ALWAYS reuse existing nodes from the Current Knowledge Graph if they capture ≥ 80% of the meaning.
– Create a new Concept Node ONLY if:
  (a) no existing node captures the conceptual essence, and
  (b) the concept is relevant to answering the Root Query.

4) New Node Output Rules
– Output ONLY newly created Concept Nodes required by your extracted edges.
– Assign new node IDs sequentially from (max existing node id + 1), format: nnumber.
– Do not create more than 3 new Concept Nodes per turn unless absolutely necessary.

5) New Edge Output Rules
– Output ALL extracted edges (they may connect to reused nodes and/or new nodes).
– Assign edge IDs sequentially from (max existing edge id + 1), format: enumber.

6) Evidence-to-Edge Mapping
– For each EN#, list ALL edge IDs that EN# directly supports.

7) Prioritization and Node Proliferation Control
– Extract only the most salient edges that advance the Root Query.
– Prefer high-level, distinctive claims (definitions, principles, stances, causal mechanisms) over minor
↪  details.
– If an EN# contains multiple ideas, keep the most representative one unless others are essential.
– Do NOT create Concept Nodes for:
  – specific examples,
  – implementation tactics,
  – temporal/contextual modifiers.
  Instead, abstract them into broader reusable concepts (principles/mechanisms/categories).

8) Abstraction Over Literalness
– A Concept Node should be generalizable beyond the single evidence instance.
– If it cannot apply in other contexts, it is likely too specific; abstract further.

# Output Requirement and Format Rules
– Output STRICTLY one valid JSON object and NOTHING else.
– JSON schema:
```

---

```
   "new_nodes": [
   "id": "nnumber", "node_name": "string", "is_core_entity": true/false
   ],
   "new_edges": [
   "id": "enumber", "source_id": "nnumber", "target_id": "nnumber", "relation_name": "string"
   ],
   "evidences_map":
   "EN1": ["enumber", "..."],
   "EN2": ["enumber", "..."]
```

## I.6. Merge Knowledge Nodes

### PROMPT TEMPLATE: MERGE KNOWLEDGE NODES

```
- You are a knowledge synthesis expert specializing in semantic concept consolidation and entity
↪  deduplication.
- Your task is to identify semantically equivalent or near-equivalent CONCEPT NODES (is_core_entity=false)
↪  in a Knowledge Graph and propose merge groups to reduce redundancy and improve semantic coherence.
- This is NOT topic discovery or unsupervised clustering; it is controlled concept normalization for graph
↪  consolidation.
- Refer to Section `Input Format` to understand the input structure.
- Refer to Section `Concept Merging Guidelines` for detailed rules.
- Refer to Section `Output Format` for the required response structure.

# Input Format
- The input consists of:
  - Root Query: topic of the deep research report.
  - Current Knowledge Graph: existing graph with nodes and edges in this exact format:

    "knowledge_nodes": [
    "node_id": "n1", "knowledge": "Node content", "is_core_entity": true/false,
    ...
    ],
    "knowledge_edges": [
    "edge_id": "e1", "representation": "Source Node - relation -> Target Node",
    ...
    ]

# Concept Merging Guidelines
1) Semantic Equivalence Threshold
- Merge concept nodes that share ≥ 85% semantic meaning AND refer to the same underlying concept in the Root
↪  Query context.
- This is semantic equivalence, not topical relatedness or surface lexical similarity.
- Example: "Virtue Ethics" + "Ethics of Character Excellence" -> merge.
- Example: "Rational Inquiry" + "Empirical Observation" -> do NOT merge.

2) Core Entity Protection Rule
- NEVER merge nodes where is_core_entity=true.
- Only concept nodes (is_core_entity=false) are eligible.

3) Representative (Canonical) Concept Naming
- For each merge group, output one representative concept name capturing the shared meaning.
- Prefer semantic precision over brevity.
- If one original node name already captures >90% of the combined meaning, reuse it unchanged.

4) Merge Group Quality Control
- Output ONLY merge groups with 2+ nodes (no singletons).
- Avoid over-merging: do NOT group nodes that are merely related or adjacent concepts.
- Maximum cluster size: 5 nodes (split if larger to maintain cohesion).

5) Justification Requirements
- Each merge group must include a concise, domain-specific justification (<50 words).
- Justify via conceptual meaning in context, not just similar wording.

# Output Format
- Output STRICTLY one valid JSON object and NOTHING else:
```

```
    "clusters": [
    "cluster_id": "c1",
    "representative_concept": "New merged concept name",
    "source_node_ids": ["n4","n5"],
    "similarity_justification": "Concise explanation (<50 words)"

    ]

- Critical rules:
  - cluster_id format: "cnumber" starting from c1.
  - source_node_ids must be strings like ["n4","n5",...].
  - Include only clusters with 2+ nodes.
  - Sort clusters by the first node ID numerically.
```

