# OpenReview forum: "A Tale of Two Graphs: Separating Knowledge Exploration from Outline Structure for Open-Ended Deep Research"
_ICML.cc/2026/Conference — ICML 2026 regular_

### Official Review · Reviewer_KHEi · 2026-03-08

**Soundness:** 3
**Presentation:** 3
**Significance:** 2
**Originality:** 2
**Overall Recommendation:** 4
**Confidence:** 4

**Summary:**

This paper introduces DualGraph, a memory architecture for deep research agents that maintains two co-evolving structures: an Outline Graph for report organization and a Knowledge Graph for storing entities, concepts, and their relations. The KG enables the agent to detect missing or weak relations through topological analysis, which then drives targeted search queries. Experiments on DeepResearch Bench, DeepResearchGym, and DeepConsult show that DualGraph achieves competitive results with proprietary systems like Gemini-2.5-Pro-DeepResearch. Ablation studies show the KG helps the agent converge faster and produce better intermediate outlines.

**Compliance With Llm Reviewing Policy:**

Affirmed.

**Key Questions For Authors:**

1.Can you ablate the individual KG components separately? Specifically, what happens with enrichment chains only, or with exploratory chains only? This would clarify whether the full topological analysis is needed.

2.Can you provide a qualitative analysis of the citation errors introduced by KG-driven exploration? Are they hallucinated, loosely relevant, or misattributed?

3.Have you tried a simpler knowledge store without community detection or SBM, just tracking entity-relation gaps? How does it compare?

4.Have you done any human evaluation, even on a small subset?

**Limitations:**

The authors discuss computational cost and LLM extraction quality, which is good. But they do not discuss the bias risks of LLM-only evaluation, the potential harm of increased but less accurate citations, or generalizability to niche domains with sparse web evidence.

**Strengths And Weaknesses:**

Strengths
1. The separation of "what the agent knows" from "how it writes" is a clean and well-motivated design. The example in Figure 1 clearly shows how outline-only methods can miss important cross-entity comparisons.
2. Evaluation covers three benchmarks with many baselines including both open-source and commercial systems. Testing with both GPT-4.1 and GPT-5 adds credibility.
3. The controlled ablation sharing the same initialization is a good choice. The per-iteration analysis in Figures 3 and 4 is informative.
4. The appendix is very detailed with full algorithms, prompts, and cost breakdowns. This helps reproducibility.

Weaknesses
1. The individual components are all well-known techniques: KG extraction, Leiden community detection, stochastic block models, outline planning. The novelty lies in combining them, but the paper does not show whether simpler alternatives could achieve similar results.
2. The only ablation is full KG vs. no KG. This is too coarse. We cannot tell which parts of the KG machinery actually matter: is it the enrichment chains, the SBM link prediction, the structural hole analysis, or just having an entity store?
3. All evaluation relies on LLM judges. DualGraph produces significantly more citations by design, and it is unclear whether GPT-4.1 as judge might systematically favor citation-heavy reports. No human evaluation is provided.
4. Citation accuracy drops when adding the KG, while citation count increases. The authors explain this as retrieving more "contextual" evidence, but this is speculation. It could also mean the system adds loosely relevant or incorrect citations.
5. Token usage increases by about 36% over the no-KG variant. The paper calls this "modest," but it is additive to an already expensive pipeline. Practical cost and latency implications are not discussed.
6. GraphRAG and HippoRAG are discussed in related work but not compared against experimentally. Since the core claim is about KG-guided exploration, comparing with other graph-based memory methods would be valuable.

---

> ### Author Rebuttal · Authors · 2026-03-31
>
> ## W1. Novelty Concern
>
> The core innovation lies in two aspects: (1) Prior deep research systems conflate what the agent knows with how it organizes the report. Knowledge is implicitly embedded in the outline structure with no independent representation. DualGraph is the first to separate these concerns: the OG manages the report outline, while the KG maintains a distinct semantic knowledge graph. This separation is what enables graph-based knowledge gap discovery, which was not possible under previous designs. (2) A graph-driven exploration mechanism. Search queries are derived from KG topology, allowing agents to systematically identify knowledge gaps and guide exploration, rather than relying on the LLM to implicitly infer gaps from the outline alone.
>
> **Baselines and ablation variants are the "simpler alternatives."** Our baselines already cover two representative categories of existing deep research approaches, which serve as simpler alternatives that do not adopt our separation design. In addition, we ablate individual components within our architecture, to verify that each is indispensable. Ablation results confirm that removing any component leads to consistent degradation (see Reviewer gy4d W5).
>
> ## W2 & Q1 & Q3 & W6. Comparison with GraphRAG-like work and DualGraph Variants
>
> See Reviewer gy4d W5 for per-iteration ablation results, including "DualGraph only EntityRelation" (a simpler store that only tracks entity-relation gaps without community detection or SBM) and "DualGraph by LightRAG", along with analysis of why RAG-oriented graph methods are not directly comparable to DualGraph's gap-detection-oriented KG.
>
> ## W3 & Q4. Concerns on LLM-as-a-Judge
>
> See Reviewer ZQB2 W1.
>
> ## W4. FACT C.acc's Single-Source Verification Underestimates Multi-source Citation Support
>
> The slight drop in FACT C.acc. should be viewed as a limitation of the evaluation protocol, not as evidence that the system is adding low-quality citations. Specifically, C.acc checks whether each citation can support the corresponding fact on its own. When a fact is synthesized from multiple citations, no single citation may fully cover the conclusion, so C.acc systematically marks it as inaccurate. DualGraph's KG encourages exactly this kind of multi-citation synthesis, making the C.acc. drop as a protocol mismatch rather than a quality decline. For example, in one case the report synthesizes three citations to form a conclusion that no single citation fully covers; C.acc marks all three as inaccurate despite the conclusion being collectively accurate.
>
> **Multi-citation joint verification experiment.** To test this hypothesis, we randomly sampled 20 cases from DeepResearchBench and conducted a joint evaluation: instead of FACT's per-citation independent verification, we let the LLM see **all citations** belong to a given fact at once before judging accuracy. We report three metrics: Multi-citation ratio (fraction of facts citing multiple citations), Avg citation/fact (average number of citations per fact), and Joint C.acc (accuracy under joint verification, directly comparable to FACT's single-source C.acc).
>
> | |DualGraph|w/o KG|
> |-|-|-|
> |Multi-citation ratio|60.7%|50.3%|
> |Avg citation/fact|2.31|1.95|
> |Joint C.acc|90.51%|90.41%|
>
> DualGraph has a higher Multi-citation ratio, confirming that KG encourages more cross citation synthesis. Yet under joint evaluation, the two systems' Joint C.acc is nearly identical, confirming that the C.acc. drop originates from FACT's single source protocol systematically penlizing multi-citation synthesis, not from a decline in citation quality.
>
> **Why DeepResearch Bench II is more appropriate.** The recently released DeepResearch Bench II（arxiv:2601.08536）does not use FACT. Instead, it employs fine-grained, expert-derived rubrics that are verifiable item by item, and evaluates systems along three dimensions. InfoRecall measures whether the report accurately and comprehensively retrieves the facts, data, and supporting evidence relevant to the task. Analysis measures whether the report can form higher-level conclusions from retrieved evidence rather than merely listing facts. These two dimensions are more appropriate for citation evaluation: broader evidence coverage and stronger cross-citation synthesis. Results are as follows:
> |Method|Overall|InfoRecall|Analysis|Presentation|
> |-|-|-|-|-|
> |DualGraph|41.48|34.21|52.45|90.65|
> |DualGraph w/o KG|39.50|32.18|50.73|90.27|
>
> With KG, InfoRecall and Analysis all improve, indicating that KG helps the system more comprehensively recover key information required by expert rubrics and better synthesize evidence into conclusions directly relevant to the task.
>
> ## W5. Token Discussion
>
> See Reviewer JZTB W1-1.
>
> ## Q2. Empirical Analysis of KG Extraction Quality
>
> See Reviewer JZTB W1-2.
>
> ## L1. Discussion Generalizability to Niche Domains
>
> This limitation is shared by all deepresearch systems, not unique to DualGraph. We will clarify that in the revision.

---

> > ### Author Rebuttal · Reviewer_KHEi · 2026-04-02
> >
> > I will change my score.

---

> > > ### Author Response · Authors · 2026-04-04
> > >
> > > Thank you very much for your thoughtful review and for taking the time to read our rebuttal carefully.
> > > We sincerely appreciate your reconsideration of the paper, and we are very glad that the additional analyses and clarifications helped address your concerns.
> > > Your comments were especially valuable in helping us strengthen the paper’s discussion of KG components, citation analysis, and evaluation limitations, and we will make sure these points are clearly reflected in the revision.

---

### Official Review · Reviewer_gy4d · 2026-03-08

**Soundness:** 3
**Presentation:** 3
**Significance:** 3
**Originality:** 3
**Overall Recommendation:** 4
**Confidence:** 5

**Summary:**

The key problem studied by this manuscript is how to improve long-horizon information exploration and report generation in Open-Ended Deep Research (OEDR) agents. The authors attempt to present a general aspect of separating knowledge representation from report structure by introducing DualGraph, a dual-memory architecture that maintains both an Outline Graph (OG) and a Knowledge Graph (KG). The OG models the structural organization of the final report, while the KG captures atomic knowledge units and their relations extracted from retrieved evidence. By jointly evolving these two graphs and using KG topology to identify knowledge gaps, the framework generates targeted search queries that guide iterative exploration. Experimental results on DeepResearch Bench, DeepResearchGym, and DeepConsult show improvements in report depth, breadth, and evidence coverage compared with existing deep research systems. Overall, the paper proposes a structured memory mechanism for long-horizon research agents and demonstrates its effectiveness through empirical evaluations.

**Compliance With Llm Reviewing Policy:**

Affirmed.

**Final Justification:**

After the rebuttal, the authors supplemented relevant references, which is important for the graph+agent field. Therefore, I maintain my score and raise my sub-scores.

**Key Questions For Authors:**

See the Weaknesses.

**Limitations:**

Yes.

**Strengths And Weaknesses:**

Strengths:
1. The paper identifies an important limitation of existing open-ended research agents, namely the conflation of knowledge representation and report structure, and clearly motivates the need for separating these two aspects to enable more systematic exploration.
2. The DualGraph framework provides a conceptually clean separation between the Outline Graph and Knowledge Graph, which improves interpretability and offers a structured way to represent both narrative organization and factual knowledge.
3. The method is evaluated on several challenging OEDR benchmarks, including DeepResearch Bench, DeepResearchGym, and DeepConsult, showing consistent improvements in report quality metrics such as depth, breadth, and evidence grounding.
4. The paper includes ablation studies comparing the full model with variants without the knowledge graph, demonstrating that KG-guided exploration leads to faster convergence and improved coverage during iterative research.

Weaknesses:
1. Nodes and relations in the KG are entirely extracted from retrieved evidence using an LLM. If the extraction process introduces noise, hallucinated relations, or semantic bias, such errors may propagate and accumulate across iterations, potentially amplifying incorrect knowledge structures and negatively affecting subsequent search strategies and exploration trajectories.
2. Several core evaluation metrics across the adopted benchmarks (e.g., RACE in DeepResearch Bench and the scoring metrics in DeepResearchGym) depend on LLM-based judges. While convenient, such evaluation schemes may suffer from model bias, instability, or alignment artifacts, raising concerns about the objectivity and reproducibility of the reported improvements.
3. DualGraph requires multiple additional operations in each iteration, including knowledge extraction, graph construction and updating, relation discovery, and graph-driven query generation. Compared with traditional outline-centric research agents, this design incurs additional LLM calls and system complexity, which may limit scalability and efficiency in real-world deployments.
4. Although the empirical results demonstrate performance improvements, the work mainly provides experimental validation. The paper does not offer formal analysis regarding how DualGraph reduces the exploration search space, improves knowledge coverage, or affects the convergence behavior of iterative research processes.
5. The paper does not sufficiently reference several important recent works in graph-enhanced retrieval and agent reasoning, such as LightRAG, Graph-R1, GraphSearch, and GraphAgent, which explore similar ideas of leveraging graph structures for retrieval, reasoning, or agent planning. Including and comparing with these works would strengthen the positioning of the proposed method and clarify its novelty relative to existing graph-based research agent frameworks.

[1] Guo, Z, et al. "Lightrag: Simple and fast retrieval-augmented generation."

[2] Luo, H, et al. "Graph-r1: Towards agentic graphrag framework via end-to-end reinforcement learning."

[3] Yang, C, et al. "GraphSearch: An Agentic Deep Searching Workflow for Graph Retrieval-Augmented Generation."

[4] Yang, Y, et al. "Graphagent: Agentic graph language assistant."

---

> ### Author Rebuttal · Authors · 2026-03-31
>
> ## W1. Empirical Analysis of KG Extraction Quality
> See Reviewer JZTB W1-2
> ## W2. LLM-as-a-Judge Concerns
> See Reviewer ZQB2 W1
> ## W3. Quantify KG Maintenance Cost
> See Reviewer JZTB W1-1
> ## W4. Formal Analysis of How DualGraph Impact Search Space, Coverage and Convergence
> Deep research must decide what to search next at each iter. Let $\mathcal{Q}_t$ denote the candidate query space at iteration $t$. Without structural constraints, $\mathcal{Q}_t$ is freely generated by the LLM and can be arbitrarily large.
>
> **Search space under KG constraints.** DualGraph decomposes $\mathcal{Q}_t$ into two bounded subsets:
>
> - Enrich space: $\mathcal{Q}\_t^{\text{enrich}} \subseteq \lbrace e \in E_t \mid \text{evidence-count}(e) \leq \tau\_{\text{enrich}} \rbrace$, i.e., existing but under-evidenced edges in the KG. Its upper bound is $|E_t|$ (current edge count), and after ranking by the composite enrichment score $\text{Score}_{\text{enrich}}(u,v) = \frac{1 + \text{node-importance}(u,v) + \text{cross-community}(u,v)}{1 + \text{evidence-count}(u,v)}$, only the top $\lfloor N\_{\text{KG}}/4 \rfloor$ are selected. This subset addresses "known but under-evidenced" gaps — strengthening existing relations.
> - Explore space: $\mathcal{Q}\_t^{\text{explore}} \subseteq (V_t \times V_t) \setminus E_t$, i.e., node pairs not yet connected in the KG. Although theoretically $O(|V_t|^2)$, triple filtering retains only top-$\lfloor N_{\text{KG}}/4 \rfloor$ candidates per type: Type I (semantic similarity) addresses "should exist but undiscovered" links; Type II (SBM cross-community probability/entropy) addresses "implicit cross-topic links"; Type III (structural hole signals) addresses "gaps between topic islands".
>
> Thus, the total number of KG-driven search chains per iteration is strictly bounded by $N_{\text{KG}}$, and the total search chain count is $N = N_{\text{OG}} + N_{\text{KG}}$.
>
> **Monotonic coverage growth and search space shrinkage.** Critically, each iteration covers new ground without redundantly re-exploring previously resolved gaps:
>
> - For Enrich, once an edge accumulates sufficient evidence ($\text{evidence-count}(e) > \tau_{\text{enrich}}$), it exits the candidate set permanently.
> - For Explore, once a node pair is investigated, the KG update either adds it to $E_{t+1}$ or reprioritizes it under the new topology, preventing redundant queries.
>
> Since newly retrieved evidence simultaneously updates the KG and OG, the set of resolved knowledge gaps is monotonically non-decreasing across iterations. The effective residual search space,  therefore shrinks over time, transforming exploration from "guessing what's missing" to "graph topology indicating what's missing."
>
> **Convergence.** The bounded search space and nonrepetitive gap resolution ensure the residual candidate set is finite and monotonically decreasing, guaranteeing convergence. Greedy selection of highest scoring gaps further concentrates gains in early iterations.
>
> ## W5. Comparing with Graph-based Methods and Variants.
>
> Graph-based methods (e.g., LightRAG) primarily leverage the KG for efficient retrieval, whereas DualGraph analyzes KG topology to identify knowledge gaps rather than retrieving from it. Although the design goals differ, we still constructed a direct comparison: DualGraph by LightRAG, which replaces our KG construction pipeline with LightRAG's implementation, while keeping the rest of DualGraph unchanged.
>
> To ensure fair comparison, we disable early stopping for all variants, and evaluate across two dimensions at each iteration:
>
> Table X: Outline Quality per Iteration
>
> ||Iter1|Iter2|Iter3|Iter4|Iter5|
> |-|-|-|-|-|-|
> |DualGraph|76.7|81.63|85.55|87.6|88.58|
> |DualGraph w/o KG|75.15|78.88|81.12|82.85|84.32|
> |DualGraph w/o Enrich|75.25|79.62|82.38|84.6|85.58|
> |DualGraph w/o Explore|75.9|80.83|83.78|85.45|86.78|
> |DualGraph only EntityRelation|75.43|79.22|81.68|83.65|84.83|
> |DualGraph by LightRAG|75.7|78.75|81.28|83.2|84.5|
>
> Table Y: Cumulative Search Query & Evidence Quality per Iteration
>
> | |Iter1|Iter2|Iter3|Iter4|Iter5|
> |-|-|-|-|-|-|
> |DualGraph (Ours)|66.04|69.91|72.58|73.44|74.37|
> |DualGraph w/o KG|62.38|65.01|67.76|69.46|70.03|
> |DualGraph w/o Enrich|64|67.41|70.42|71.31|71.67|
> |DualGraph w/o Explore|64.78|68.33|71.21|72.32|73.05|
> |DualGraph only EntityRelation|63.58|66.98|69.63|70.92|71.26|
> |DualGraph by LightRAG|64.34|66.08|68.63|70.23|70.79|
>
> It shows that DualGraph consistently outperforms all variants across all iterations and both metrics. Removing either Enrich or Explore chains leads to performance degradation, confirming the indispensable roles of the two chain types. DualGraph only EntityRelation also underperforms, confirming that topological signals beyond raw entity-relation gaps are essential. DualGraph by LightRAG performs significantly worse than the full DualGraph, suggesting its KG construction is not suited for gap detection (producing overly fine-grained, numerous triplets that hinder effective topological gap analysis).

---

> > ### Author Rebuttal · Reviewer_gy4d · 2026-04-03
> >
> > The authors have partially addressed my concern. Since this paper is a graph + agent work, please clarify the differences and performance comparisons with the other works besides LightRAG, which are not mentioned in the paper, such as Graph-r1, GraphSearch, and Graphagent.

---

> > > ### Author Response · Authors · 2026-04-04
> > >
> > > Thanks for the follow-up. We checked all papers and think the differences are worth explaining in detail.
> > >
> > > **Difference in KG purpose and task setting.** While LightRAG, Graph-R1, GraphSearch, and GraphAgent all involve graph structures, their KG purpose and task setting differ from DualGraph in important ways:
> > >
> > > | System               | KG Purpose                                                   | Task                                                   | Benchmarks                                            |
> > > | -------------------- | ------------------------------------------------------------ | ------------------------------------------------------ | ----------------------------------------------------- |
> > > | LightRAG [1]         | **Retrieval index**: entity-relation graph over a static document corpus for efficient retrieval | Multi-hop document QA                                  | UltraDomain Benchmark                                 |
> > > | Graph-R1 [2]         | **Retrieval index**: retrieve relevant facts to answer a given question | Multi-hop document QA                                  | HotpotQA, MuSiQue, 2WikiMultiHop, NQ, PopQA, TriviaQA |
> > > | GraphSearch [3]      | **Retrieval index**: dual-channel retrieval over text chunks and graph edges | Multi-hop document QA                                  | HotpotQA, MuSiQue, 2WikiMultiHop                      |
> > > | GraphAgent [4]       | **Reasoning substrate**: LLM reasons over a static document graph for classification/generation | Node classification; Text summarization and generation | IMDB, ACM, ArXiv-Papers,GovReport Summarize           |
> > > | **DualGraph (Ours)** | **Gap detector**: KG topology identifies *what knowledge is missing* and drives the next web search. The KG is not queried for retrieval. | Open-ended long-form research                          | DeepResearch Bench, DeepResearchGym, DeepConsult      |
> > >
> > > LightRAG, Graph-R1, and GraphSearch are GraphRAG systems where the graph is a retrieval substrate for QA over a closed, static document corpus. GraphAgent applies LLMs over fixed document graphs for classification, generation and summarization. In all four cases, the graph is essentially a store of existing knowledge that the system retrieves from.
> > >
> > > In DualGraph, the KG is used differently: it tracks entities, concepts, and relations extracted from retrieved evidence, and its **topology** is analyzed to find missing or weakly supported knowledge, which drives the next round of web search. The KG is never retrieved from directly. This is paired with a separate Outline Graph for report organization, a design that none of these works adopt.
> > >
> > > **Fundamental difference in data assumption.** Beyond the task-level difference, there is a deeper architectural incompatibility. LightRAG, Graph-R1, GraphSearch, and GraphAgent all assume a closed, static document corpus that is fully ingested into the KG during an offline preprocessing stage. At query time, the graph is fixed and serves solely as a retrieval index. DualGraph operates under a fundamentally different assumption: there is no pre-existing corpus. The KG starts empty and is built incrementally as the agent retrieves new evidence from the open web at each iteration. This means the graph construction pipelines of these systems cannot be meaningfully transplanted into DualGraph's online, incremental setting, because they are designed to index a complete corpus, not to grow from a stream of newly discovered documents.
> > >
> > > **On the LightRAG controlled comparison.** In our earlier rebuttal, we ran a controlled variant called "DualGraph by LightRAG" (refer to rebuttal on JZTB W5), which swaps out only our KG construction pipeline with LightRAG's while keeping everything else the same. LightRAG is also not an OEDR system, but this swap let us test whether a retrieval-oriented KG construction could substitute for our gap-detection-oriented one. It could not: performance dropped clearly across all iterations. Running a similar substitution for Graph-R1, GraphSearch, or GraphAgent is harder to set up cleanly given their different task formulations and evaluation settings.
> > >
> > > We will discuss all these works in the related work section of the revision. We hope these clarifications help address the remaining concern, and we would appreciate the reviewer's reconsideration of the score if they find the distinctions convincing.
> > >
> > > **References**
> > >
> > > [1] Guo, Z, et al. "LightRAG: Simple and fast retrieval-augmented generation."
> > >
> > > [2] Luo, H, et al. "Graph-R1: Towards agentic graphrag framework via end-to-end reinforcement learning."
> > >
> > > [3] Yang, C, et al. "GraphSearch: An Agentic Deep Searching Workflow for Graph Retrieval-Augmented Generation."
> > >
> > > [4] Yang, Y, et al. "GraphAgent: Agentic graph language assistant."

---

### Official Review · Reviewer_ZQB2 · 2026-03-09

**Soundness:** 3
**Presentation:** 4
**Significance:** 3
**Originality:** 3
**Overall Recommendation:** 5
**Confidence:** 4

**Summary:**

The paper presents DualGraph, an approach for improving LLMs' ability to perform deep research tasks. The core idea of DualGraph is to harmonize outline-centric agentic approaches for deep research with a knowledge graph which stores the current state of knowledge and guides further search and exploration. The approach thus maintains an "outline graph" and "knowledge graph", which together help to more effectively guide the search process. Experiments are performed on 3 deep research benchmarks, demonstrating strong performance against a variety of proprietary deep research systems.

**Compliance With Llm Reviewing Policy:**

Affirmed.

**Final Justification:**

I will maintain my accept recommendation after author rebuttal and considering other reviewer opinions. I believe it makes worthwhile contributions and provides interesting ideas that future work can further build upon.

**Key Questions For Authors:**

Q1. Appendix D shows cost comparison of DualGraph with or without KG. Do you have any sense of how DualGraph compares to other open-source deep research approaches, in terms of time taken and input/output token cost? How much might the increased performance be a tradeoff of higher compute cost?

Q2. Fig4 shows that DualGraph w KG is judged as "better" significantly more often than w/o KG after several iterations. However, evaluation results show that the difference in performance on benchmarks is quite small. Are there any insights surrounding the approach focusing on just the outline graph and its updates? Any ablation study surrounding that part of the work would be interesting to see.

**Limitations:**

yes

**Strengths And Weaknesses:**

S1. The paper is well written and easy to follow, and design decisions seem well motivated.

S2. Detailed information about the approach such as prompts are thoroughly documented in the supplemental material.

S3. Analysis such as ablation studies and cost comparison are helpful. Many of the questions that I had while reading the paper were answered in the analysis.


W1. Evaluations all rely on LLM-as-a-Judge, which is common practice and applied fairly throughout baseline comparisons but nonetheless leaves some question on exactly how well the detailed differences between various methods can really be surfaced.

W2. Results are not necessarily state of the art across all evaluations - Gemini-2.5-pro-deepresearch has best performance for DeepResearchGym and highest Avg score DeepConsult. DualGraph does show second-best performance on these tasks though (and a higher Win% on DepConsult), and the approach has notably better performance than other open-source deep research methods which use the same model (GPT 4.1).

---

> ### Author Rebuttal · Authors · 2026-03-29
>
> ## W1.Concern about Evaluations rely on LLM-as-a-Judge.
>
> We appreciate the concern on the evaualtion using LLM-as-a-Judge. First, we want to clarify that the benchmarks we adopt were designed with judge consistency in mind. For example, DeepResearch Bench uses manually reviewed rubrics and includes a dedicated human agreement study, with results showing that LLM judge rankings remain stable across different judge models, indicating limited model-specific bias.
>
> To further validate the reliability of LLM-based evaluation, we conducted another fine-grained human evaluation. We randomly sampled 5 evaluation cases each from OpenAI Deep Research and DualGraph on DeepResearch Bench, yielding 10 cases covering 254 rubric criteria. Each case includes the evaluated report, the reference benchmark report, and the LLM judge's per-criterion analysis and scores across four dimensions. Five human annotators with research experience independently reviewed each rubric criterion. After reading both reports (anonymized), annotators were instructed to mark each LLM-assigned score and its corresponding reasoning as "reasonable" or "unreasonable".
>
> The annotation results can be summarized as follows: the all-annotators-agree rate, defined as the proportion of criteria where all five annotators unanimously endorsed the LLM's score and reasoning, is 88.6% (225/254); (2) the per-judgment agreement rate, defined as the fraction of individual human judgments (each annotator × each criterion) rated as reasonable, is 97.5% (1,238/1,270); (3) inter-annotator agreement is high, with pairwise agreement at 94.3% and Gwet's AC1 at 0.95. Together, these results provide strong evidence that LLM-as-a-Judge produces reliable and consistent evaluations.
>
> ## W2. Concern about Comparison with Frontier Proprietary Systems
> Our originally submitted results on DeepResearchGym and DeepConsult used only GPT-4.1 as the backbone. After upgrading to GPT-5.2 and running on all three benchmarks, DualGraph consistently outperforms Gemini-2.5-Pro-DeepResearch:
>
> ||DeepResearch Bench RACE|DeepResearchGym Avg.|DeepConsult Avg. Score|
> |-|-|-|-|
> |Gemini-2.5-Pro-DeepResearch|52.54|96.02|6.70|
> |DualGraph (GPT-5.2)|53.08|96.52|6.84|
>
> That said, we respectfully note that DualGraph already achieves state-of-the-art performance among all open-source frameworks across all three benchmarks. Gemini-2.5-Pro-DeepResearch is a closed-source commercial product with undisclosed architecture and compute budget, making direct comparison inherently challenging.
>
> ## Q1. Time/Token Comparison with Other Open-Source Approaches
> See Reviewer JZTB W1-1
> ## Q2. Understanding the Relationship between Per-Iteration Outline Graph Advantage and Final Report Performance
> We appreciate the reviewer's observation and understand that Fig. 4 may give the impression that the per-iteration KG advantage does not fully translate to final scores. However, we want to clarify that the primary reason is the early stopping mechanism.
>
> In the benchmark result tables, each method's report is produced under the early stopping mechanism, i.e., DualGraph converges faster (avg. 3.20 iters vs. 3.81) and stops earlier, so its final report is generated at a lower iteration count. In contrast, Fig. 4's pairwise comparison is conducted at the **same iteration** for both methods on each case, which neutralizes the effect of early stopping and captures the full per-iteration KG advantage.
>
> To verify, we compared each method's **final OG** (where iteration counts may differ due to early stopping). The win rate of DualGraph vs. w/o KG drops to 61.67%, which is smaller than Fig. 4's gap — consistent with the early stopping explanation:
>
> | |Win Rate|Lose Rate|
> |-|-|-|
> |DualGraph|61.67%|38.33%|
>
> Additionally, the two evaluations measure different things with different purposes. The benchmark result tables report multi-dimensional scores on the **final report**, reflecting overall report quality. Fig. 4 reports OG-level **win/lose ratios**, reflecting how often DualGraph's OG is preferred over the w/o KG variant at matched iterations.

---

> > ### Author Rebuttal · Reviewer_ZQB2 · 2026-04-01
> >
> > Q1 was the main question I was interested in getting insight on, and this was answered well in the response to Reviewer JZTB. Q2 was clarified.

---

> > > ### Author Response · Authors · 2026-04-04
> > >
> > > Thank you very much for your thoughtful review and for taking the time to read our rebuttal. We are glad that the additional clarification addressed your concerns, especially regarding the cost comparison and the interpretation of the KG advantage. We also sincerely appreciate your positive assessment of the paper’s clarity, analysis, and overall design. Your feedback has been very helpful, and we will incorporate these clarified points into the final revision.

---

### Official Review · Reviewer_JZTB · 2026-03-11

**Soundness:** 3
**Presentation:** 4
**Significance:** 3
**Originality:** 3
**Overall Recommendation:** 4
**Confidence:** 3

**Summary:**

This paper proposes a framework for open-ended deep research agents that separates knowledge representation from report organization named DualGraph. The system maintains two co-evolving structures: an Outline Graph that represents the hierarchical structure of the report and a Knowledge Graph that stores entities, concepts, and relations extracted from retrieved evidence. By analyzing the KG, the agent can identify missing or weakly supported knowledge and generate targeted search queries to guide further exploration. Experiments on several deep research benchmarks show that this dual-graph design improves report quality, coverage, and evidence grounding compared with outline-only approaches and other baselines.

**Compliance With Llm Reviewing Policy:**

Affirmed.

**Key Questions For Authors:**

The paper reports that introducing the KG improves effective citations, but it does not disentangle which components drive this gain. Although the framework distinguishes between Enrich chains for strengthening evidence on existing relations and Explore chains for hypothesized missing relations, the evaluation only reports aggregate citation improvements. It therefore remains unclear how much of the gain comes from reinforcing existing relations, discovering genuinely new ones, or from KG-derived versus OG-derived supplementary queries.

**Limitations:**

A more detailed attribution analysis (e.g., separating citation gains from Enrich vs. Explore queries or measuring the proportion of newly discovered relations) would strengthen the paper’s claims about the benefits of KG-driven retrieval.

**Strengths And Weaknesses:**

**Strengths: **
- The dual-graph design provides a clear and principled separation between knowledge representation and report structure, enabling more explicit reasoning about information coverage and improving the interpretability of the system’s planning and retrieval process.

- The retrieval strategy is mechanism-driven rather than purely prompt-based, using structured signals from the graph to guide evidence strengthening and gap exploration, which makes the retrieval process more systematic and targeted.

** Weaknesses: **
Maintaining and updating the knowledge graph requires additional LLM calls and structured extraction steps. Moreover, errors in entity or relation extraction may propagate through the iterative retrieval process, potentially affecting subsequent planning and search decisions. The paper acknowledges this issue but does not provide a detailed analysis of its practical impact.

---

> ### Author Rebuttal · Authors · 2026-03-31
>
> ## W1-1 Token and Time Cost Analysis
> To quantify the cost of KG maintenance, we compare DualGraph with open-source baselines on a 20-case subset of DeepResearch Bench.
>
> Table1: Token Usage and Runtime Comparison
>
> |Method|Input Tokens (K)|Output Tokens (K)|Total Tokens (K)|Avg. Time (min)|RACE|
> |-|-|-|-|-|-|
> |DualGraph|2,301|189|2,490|36.75|51.5|
> |DualGraph w/o KG|1,661|137|1,798|30.68|49.18|
> |WebWeaver|1,404|132|1,536|28.8|45.2|
> |Langchain-DR|833|64|897|13.4|42.92|
> |Tongyi-DR|335|15|350|7.1|40.95|
> |Qwen-Agent|425|21|446|9.7|40.28|
>
> Table2: Comparison Per-Module Breakdown
>
> |Module|DualGraph Tokens (K)|DualGraph Time (min)|w/o KG Tokens (K)|w/o KG Time (min)|
> |-|-|-|-|-|
> |Generate Queries|48|0.43|26|0.27|
> |Judge Terminal|76|1.01|79|1.39|
> |Maintain Outline|127|2.08|96|1.97|
> |Maintain KG|224|3.48|—|—|
> |WebSearch/Crawl|1,673|26.08|1,308|23.46|
> |Write Paper|341|3.67|289|3.59|
> |Total|2,490|36.75|1,798|30.68|
>
> From Table 1, we acknowledge that DualGraph incurs additional computational cost compared to other open-source methods. However, this cost enables our system’s multiple modules to generate better search queries, perform more comprehensive exploration, and collect more complete information sources. This design emphasizes understanding query directions, which is the core driver of our system’s superior performance. Moreover, DualGraph achieves the highest RACE, demonstrating that the additional cost leads to meaningful performance gains.
>
> From Table 2, KG maintenance accounts for only 9% of total tokens, while more comprehensive web search and crawling dominates at 67%, which is an unavoidable cost for high-quality deep research. This shows that DualGraph's overhead is primarily driven by more comprehensive exploration, not by the KG component itself.
>
> Ultimately, in the report generation domain, the primary goal is to deliver high-quality, trustworthy, and verifiable reports. Our current design prioritizes this high standard.
>
> ## W1-2 Low Error Rate and Iterative Self-Correction Mitigate KG Noise
> We acknowledge that LLM-based extraction of graph entities and relations inevitably introduces errors. We analyzed all 6,014 entity-relation triplets extracted at Iter 1 across 100 cases in DeepResearch Bench: only 2.4% are anomalous, consistent with recent studies (GenRES (Jiang et al., NAACL 2024) showing high accuracy of LLMs on structured information extraction tasks. The anomaly details are as follows:
>
> |Category|% of All Edges|Definition|
> |-|-|-|
> |Loosely Relevant|1.48%|The extracted relation is related but not directly stated in the evidence|
> |Hallucination|0.65%|The triplet has no grounding in the evidence|
> |Semantic Bias|0.27%|The relation direction or meaning is subtly distorted|
> |Total|2.40%||
>
> Only **0.12%** (7 triplets) of all triplets are both anomalously extracted and used to drive search queries before correction. In contrast, most of the remaining anomalous extractions are mitigated by DualGraph’s built-in self-correction mechanism. Specifically:
>
> 1.Although *Loosely Relevant* extractions are not directly supported in the current iteration, they serve two purposes: helping generate new search queries and, in many cases, being later validated by newly retrieved evidence confirming the triplet is factually correct.
>
> 2.DualGraph performs merge or update operations on the KG at each iteration(see Methods §3.3), which self-corrects errors including *Hallucination* and *Semantic Bias*. If newly retrieved evidence contradicts an existing relation, the system automatically corrects or deletes the erroneous edge. For example, in Case 79, the initial extraction was "(Misrepresentation of LGBTQ+ in MENA media)-[exacerbates]->(Legal barriers for transgender people)", which reverses the true causality (*Semantic Bias*). After retrieving new evidence, the system automatically corrected the relation to "-[results from]->".
>
> We will incorporate the above discussion in the revision to further clarify the robustness of the approach.
>
>
> ## Q1&L1. Analysis about Citation Gains from Enrich vs. Explore Queries
>
> We traced the source of each effective citation produced by DualGraph on DeepResearch Bench, recording which query type ultimately retrieved it. The results are as follows:
>
> |Citation Source|Proportion|
> |-|-|
> |OG-derived queries|58.0%|
> |KG-derived queries|42.0%|
> |└ KG-Enrich|26.9%|
> |└ KG-Explore|15.1%|
>
> This demonstrating that the KG plays an indispensable role in evidence discovery. And this shows that the KG's gains do not stem solely from discovering new knowledge, but also from reinforcing existing but insufficiently evidenced relations. This highlights the value of KG topological design, where outline centric systems cannot explicitly identify which relations lack sufficient evidence support. We will include this citation breakdown analysis in the revision.

---

> > ### Author Rebuttal · Reviewer_JZTB · 2026-04-04
> >
> > The authors have addressed my concerns.

---

> > > ### Author Response · Authors · 2026-04-04
> > >
> > > Thank you very much for your careful reading of our rebuttal and for your positive acknowledgement. We are glad that the additional analyses have adequately addressed your concerns. We will incorporate these analyses and clarifications into the final revision to further improve the paper’s completeness and transparency.
> > > Thank you again for your constructive feedback.

---

### Decision · Program_Chairs · 2026-04-30

**Decision:**

Accept (regular)

**Comment:**

This paper proposes DualGraph, a framework for open-ended deep research agents that separates knowledge representation from report organization by maintaining a Knowledge Graph and an Outline Graph. The main idea is that this separation helps the agent identify missing or weakly supported knowledge more explicitly and issue more targeted follow-up searches, improving long-horizon research and report generation.

Reviewers generally found the paper clear, well motivated, and technically solid. They especially valued the graph-guided retrieval strategy and the strong empirical coverage across multiple benchmarks. The main concerns were the added cost and complexity of maintaining the KG, the possibility of error propagation from LLM-based extraction, the reliance on LLM-as-a-Judge evaluation, and the need for clearer attribution of which KG components drive the gains.

The rebuttal addressed most of these concerns effectively by adding analyses on runtime/token cost, extraction noise, and comparisons to simplified variants and related graph-based methods. Several reviewers explicitly stated that their concerns were resolved or largely resolved after discussion. The remaining issues are best viewed as limitations to clarify rather than reasons to reject the paper.

Overall, the consensus is positive. For the final version, the authors should incorporate the rebuttal evidence more directly into the paper, especially on component-level attribution, computational tradeoffs, related-work positioning, and the limitations of LLM-based evaluation.